# Understanding the Difficulties of Posterior Predictive Estimation

**Abhinav Agrawal** [1]  **Justin Domke** [1]

## Abstract

Predictive posterior densities (PPDs) are essential in approximate inference for quantifying predictive uncertainty and comparing inference methods. Typically, PPDs are estimated by simple Monte Carlo (MC) averages. In this paper, we expose a critical under-recognized issue: the signal-to-noise ratio (SNR) of the simple MC estimator can sometimes be extremely low, leading to unreliable estimates. Our main contribution is a theoretical analysis demonstrating that even with exact inference, SNR can decay rapidly with an increase in (a) the mismatch between training and test data, (b) the dimensionality of the latent space, or (c) the size of test data relative to training data. Through several examples, we empirically verify these claims and show that these factors indeed lead to poor SNR and unreliable PPD estimates (sometimes, estimates are off by hundreds of nats even with a million samples). While not the primary focus, we also explore an adaptive importance sampling approach as an illustrative way to mitigate the problem, where we learn the proposal distribution by maximizing a variational proxy to the SNR. Taken together, our findings highlight an important challenge and provide essential insights for reliable estimation.

## 1. Introduction

Given a model with prior $p(z)$ and likelihood $p(\mathcal{D}|z)$, training data $\mathcal{D}$, and an approximate posterior $q_{\mathcal{D}}(z)$, the predictive posterior density (PPD) of some test data set $\mathcal{D}^*$ under $q_{\mathcal{D}}$ is defined as

$$\text{PPD}_q := \int p(\mathcal{D}^*|z) q_{\mathcal{D}}(z) dz. \tag{1}$$

---

[1]Manning College of Information and Computer Sciences, University of Massachusetts, Amherst, MA, USA. Correspondence to: Abhinav Agrawal <aagrawal@cs.umass.edu>.

*Proceedings of the 42nd International Conference on Machine Learning*, Vancouver, Canada. PMLR 267, 2025. Copyright 2025 by the author(s).

$\text{PPD}_q$ is used extensively in approximate inference for making predictions and forecasts (Filos et al., 2019; Kompa et al., 2021; Immer et al., 2021; Petropoulos et al., 2022; Tyralis & Papacharalampous, 2022; Martin et al., 2024) and evaluating inference methods where higher $\text{PPD}_q$ values indicate better results (Wu et al., 2017; Wenzel et al., 2020; Izmailov et al., 2021b; Kim et al., 2022; Reichelt et al., 2022b; Yu & Zhang, 2023; Zimmermann et al., 2023; Sendera et al., 2024; Cheng et al., 2024). (Note, $\text{PPD}_q$ in eq. (1) is different from the PPD under the true posterior, see section A for a discussion.) The integral in eq. (1) is typically estimated via the simple Monte Carlo (MC) estimator

$$R_K = \frac{1}{K} \sum_{k=1}^{K} p(\mathcal{D}^*|z_k) \quad \text{where} \quad z_k \sim q_{\mathcal{D}}. \tag{2}$$

It is typical to work in log-space and use $\log R_K$ to estimate $\log \text{PPD}_q$. From Jensen's inequality, we know $\log R_K$ is a biased estimator of $\log \text{PPD}_q$ (with a bias related to $R_K$'s variance, see section A). This paper is motivated by the observation that this bias can be extremely large, leading to unreliable $\text{PPD}_q$ estimates.

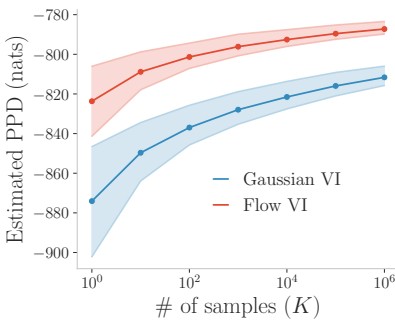

Figure 1: **Unreliable $\text{PPD}_q$ estimates.** $\log \text{PPD}_q$ estimates for a user-preference model on the MovieLens-25M dataset (Harper & Konstan, 2015), with approximate posterior $q_{\mathcal{D}}$ produced from variational inference (VI) with either a Gaussian or flow-based family. Lines show the mean $\log R_K$ and shaded regions show 95% intervals (uses 1000 repetitions.) Even with a million samples ($K = 10^6$), we do not appear to have accurate estimates.

Figure 1 shows $\log \text{PPD}_q$ estimates for a user-preference model on the MovieLens-25M dataset (Harper & Konstan, 2015), with approximate posterior $q_{\mathcal{D}}$ produced from variational inference (VI) with either a Gaussian or flow-based

family (see section 5.4 for setup). Even with a million samples, the curves are still increasing, indicating bias remains and comparisons between the two VI methods are unreliable. Such failures raise the question: *Why and when is simple MC unreliable for $PPD_q$ estimation?*

The main contribution of this paper is to develop a better understanding of the $PPD_q$ estimation problem. To do this, we focus our analysis on the signal-to-noise ratio (SNR) of $R_K$ defined as $\text{SNR}(R_K) = \mathbb{E}[R_K]/\sqrt{\mathbb{V}[R_K]}$. This definition captures how high or low is the variance relative to the target value, and becomes crucial when the target value is small. SNR has been used in the past to study gradient estimators when gradient values can be extremely small (Roberts & Tedrake, 2008; Rainforth et al., 2018; Tucker et al., 2019; Finke & Thiery, 2019; Liévin et al., 2020; Geffner & Domke, 2021; Rudner et al., 2021). Since $PPD_q$ values are (generally) numerically small, we focus on studying SNR (see section A for more discussion),

In section 2, we first provide a general expression for the SNR in any model assuming exact inference. Our subsequent analyses (eq. (5), theorem 2.2, and corollary 2.3) finds that SNR decays quickly with increase in (1) the degree of "mismatch" between the test and training data, (2) the dimensionality of the latent variable space, and (3) the size of the test data relative to training data. Several illustrative examples (figs. 4 to 6) empirically validate these findings (see fig. 2 for demonstration on a linear regression model). In section 3, we also extend our analysis to approximate inference where $q_\mathcal{D} \neq p(z|\mathcal{D})$.

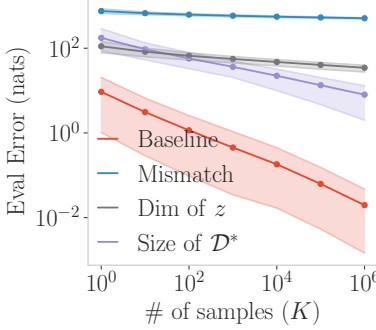

Figure 2: **Error worsens with the three factors.** $PPD_q$ estimation error for a linear regression model. For baseline, none of the three factors influencing the SNR are high and the estimation error is low. Error increases dramatically when either of three factors is high (see section 5.2 for setup).

We also explore the use of adaptive importance sampling (IS) as an illustrative way to resolve poor estimation, where we learn a parameterized proposal by optimizing a variational proxy to the SNR (section 4). This strategy, referred to as learned IS, vastly improves estimates, including a five-fold improvement in the estimated difference between the VI methods from fig. 1 (see table 4).

## 2. Analysis With Exact Inference

This section focuses on understanding the SNR of the naive estimator from eq. (2) under exact inference; that is, $q_\mathcal{D}(z) = p(z|\mathcal{D})$. Working with exact inference removes the confounding effects of the posterior approximations. We start by deriving SNR expressions in any model and then analyze these expressions to identify the influential factors.

Let $\mathcal{D} = \{y_1, y_2, \ldots, y_{|\mathcal{D}|}\}$ be the training data, $\mathcal{D}^* = \{y_1^*, y_2^*, \ldots, y_{|\mathcal{D}^*|}^*\}$ be a test dataset, where $y$ can be discrete, continuous, or mixed. Then, the following result gives two equivalent expressions for $\text{SNR}(R_K)$. (Note: We use multiset notation, so $\mathcal{D} + \mathcal{D}^*$ is the multiset addition with $2\mathcal{D} = \mathcal{D} + \mathcal{D}$ (Costa, 2021)).

**Theorem 2.1.** *Let $R_K$ be as in eq. (2) with exact inference; that is, $q_\mathcal{D}(z) = p(z|\mathcal{D})$. Let $p(z, \mathcal{D}) = p(z)\prod_{y\in\mathcal{D}}p(y|z)$. Then, $\text{SNR}(R_K) = \sqrt{K}/\sqrt{\exp(\delta)^2 - 1}$ for*

$$\delta = \frac{1}{2}KL\left(p(z|\mathcal{D} + \mathcal{D}^*) \parallel p(z|\mathcal{D})\right)$$
$$+ \frac{1}{2}KL\left(p(z|\mathcal{D} + \mathcal{D}^*) \parallel p(z|\mathcal{D} + 2\mathcal{D}^*)\right) \tag{3}$$
$$= \frac{V(\mathcal{D}) + V(\mathcal{D} + 2\mathcal{D}^*)}{2} - V(\mathcal{D} + \mathcal{D}^*) \tag{4}$$

*where $V$ is the log-normalization function $V(\mathcal{D}) = \log\int p(\mathcal{D}|z)p(z)dz$.*

*Proof sketch.* A simple calculation gives $\text{SNR}(R_1) = \sqrt{K}/\sqrt{\exp(\delta)^2 - 1}$ where $\delta := \frac{1}{2}\log\mathbb{E}[R_1^2] - \log\mathbb{E}[R_1]^2$ (lemma C.1). Using $p(\mathcal{D}|z) = \prod_{y\in\mathcal{D}}p(y|z)$ and simple algebra, gives $\mathbb{E}[R_1^c] = \exp V(\mathcal{D} + c\mathcal{D}^*)/\exp V(\mathcal{D})$ for all non-negative integers $c$ and $V(\mathcal{D}) = \log\int p(\mathcal{D}|z)p(z)dz$ (lemma C.3). Plugging $\mathbb{E}[R_1^c]$ with $c = 1$ and $c = 2$ in $\delta$, and simplifying gives eq. (4). Then, some simple observations give an expression for KL-divergence between two posteriors in terms of likelihood ratios and log-normalization constants (see lemma C.4). Applying this to each KL divergence in eq. (3) and averaging gives eq. (4). (For a formal proof, see section C.) $\square$

To understand this result, note that if $\delta$ is reasonably large, then $\text{SNR}(R_K) \approx \sqrt{K}\exp(-\delta)$ (see fig. 3). So, we can focus on $\delta$ to understand SNR. The rest of this section analyzes $\delta$ as in eq. (3) to determine the influential factors.

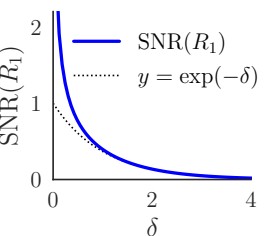

Figure 3: SNR rapidly decays with $\delta$.

KL-divergences in eq. (3) show that SNR is determined by how different the posterior $p(z|\mathcal{D} + \mathcal{D}^*)$ is from the posteriors $p(z|\mathcal{D})$ and

$p(z|\mathcal{D} + 2\mathcal{D}^*)$. This gives us the first factor: mismatch between training and test data. Intuitively, if datasets mismatch such that adding or subtracting $\mathcal{D}^*$ significantly changes the posterior $p(z|\mathcal{D} + \mathcal{D}^*)$, then the SNR will be small.

It is important to also analyze the case when datasets match; that is, when adding or subtracting $\mathcal{D}^*$ would *not* have a significant effect on $p(z|\mathcal{D} + \mathcal{D}^*)$. Intuitively, if the training and test data are similar and large such that the posteriors involving $\mathcal{D}^*$ are concentrated similar to $p(z|\mathcal{D})$, then the divergence in eq. (3) will be small. To understand what remaining factors influence SNR in such conditions, we make some simplifying assumptions and provide an informal result. Assume the following hold.

A1: The datasets $\mathcal{D}^*$ and $\mathcal{D}$ are large enough that the KL-divergence between the posteriors in eq. (4) is well-approximated by the KL-divergence between the corresponding Bayesian CLT approximations.

A2: The datasets $\mathcal{D}$, $\mathcal{D} + \mathcal{D}^*$, and $\mathcal{D} + 2\mathcal{D}^*$ are similar enough that MLE and Hessian of the *average* log-likelihood are the same for all three.

Then, if $d$ is the number of dimensions of $z$, we have

$$\delta \approx \frac{d}{2} \log \frac{1 + |\mathcal{D}^*| / |\mathcal{D}|}{\sqrt{1 + 2|\mathcal{D}^*| / |\mathcal{D}|}}. \tag{5}$$

In section D, we provide a detailed version of this result (alongside the discussion for why we need A1 than simply applying Bayesian CLT). Here, we focus on the implication of eq. (5). The right-hand-side of the eq. (5) is well approximated by $\frac{d}{4} \log (|\mathcal{D}^*|/|\mathcal{D}|)$ when $|\mathcal{D}^*|/|\mathcal{D}|$ is large. Therefore, when datasets match, the SNR still depends on dimensionality of the latent space and the relative sizes of the datasets: $\delta$ increases linearly in the number of dimensions and logarithmically in $|\mathcal{D}^*| / |\mathcal{D}|$. This completes identification of the three factors that influence the SNR: dataset mismatch, dimensionality, and relative size of $\mathcal{D}^*$.

## 2.1. Example: Gaussian Linear Regression

While theorem 2.1 provides a general expression for SNR, it required some intuition to see the impact of the three factors. With the aim to make these relationships more explicit, this section considers a linear regression model with analytically tractable Gaussian posteriors. The closed-form $\delta$ expression for this model also remain terse and we make some simplifying assumptions (like, $\mathcal{D}^*$ contains copies of training data) to gather more intuition. The following result provides a single expression capturing the the impact of the three factors on SNR.

**Theorem 2.2.** *Let* $p(z) = \mathcal{N}(z|\mu_0, \Sigma_0)$ *and* $p(y_\mathcal{D}|z) = \mathcal{N}(y_\mathcal{D}|X_\mathcal{D}z, \sigma^2 I)$, *where* $y_\mathcal{D} \in \mathbb{R}^{|\mathcal{D}|}$ *are the responses,* $X_\mathcal{D} \in \mathbb{R}^{|\mathcal{D}| \times d}$ *are the features,* $z \in \mathbb{R}^d$ *are the weights, and* $\sigma^2$ *is the known variance. Let* $\delta$ *be as in* eq. (3).

B1: *The test features* $X_{\mathcal{D}*}$ *consist of* $m$ *copies of* $X_\mathcal{D}$ *and responses* $y_{\mathcal{D}*}$ *consist of* $m$ *copies of* $y_\mathcal{D} + \Delta$ *where* $\Delta$ *is the mismatch vector of size* $|\mathcal{D}|$.

B2: *For a sequence of increasingly large training datasets, posterior is dominated by data. Mathematically, as* $|\mathcal{D}| \to \infty$, $(X_\mathcal{D}^\top X_\mathcal{D})^{-1} \Sigma_0^{-1} \to 0$.

*Assume B1 and B2. Then,* $\lim_{|\mathcal{D}| \to \infty} |\delta - \delta_{\lim}| = 0$ *and*

$$\delta_{\lim} = \frac{d}{2} \log \frac{1 + m}{\sqrt{1 + 2m}}$$
$$+ \frac{1}{2\sigma^2} \frac{m^2}{2m^2 + 3m + 1} \Delta^\top X_\mathcal{D} \left(X_\mathcal{D}^\top X_\mathcal{D}\right)^{-1} X_\mathcal{D}^\top \Delta. \tag{6}$$

We discuss the proof for this result in section E. B1 is necessary to simplify the expressions and B2 essentially means the posteriors in eq. (3) are dominated by data. This is analogous to (but weaker than) A1. In fact, if there is no dataset mismatch ($\Delta = 0$), then the right-hand side of eq. (6) reduces exactly to that of eq. (5).

The expressions in eq. (6) explicitly combine the three factors: mismatch ($\Delta$), dimensionality ($d$), and relative size of $\mathcal{D}^*$ ($m$). With some manipulation, the following bounds hold that bring out these relationships even more clearly (see section E.1 for derivation of the bounds).

$$\frac{d}{4} \log \frac{m}{2} \leq \delta_{\lim} \leq \frac{d}{4} \log \left(\frac{m}{2} + 1\right) + \frac{1}{4\sigma^2} ||\Delta||_2^2. \tag{7}$$

Overall, eqs. (6) and (7) capture the strength of the three factors: $\delta$ is affected quadratically by mismatch, linearly by the dimensionality, and logarithmically by the relative size.

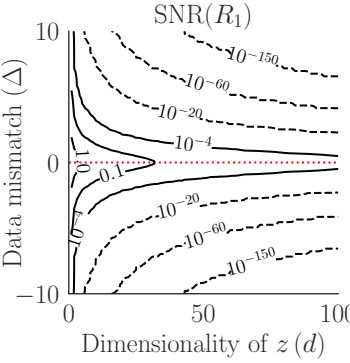

Figure 4: **SNR isocontours.** SNR for the Gaussian linear regression model where the mismatch $\Delta$ and the dimensionality of $z$ are varied, keeping relative size of $\mathcal{D}^*$ fixed ($m = 1$).

We consider an instance of the linear regression model from theorem 2.2 in figs. 2 and 4. Figure 4 shows SNR decays quickly with an increase in mismatch and dimensionality. Figure 2 shows the average difference between $\log \text{PPD}_q$

and $\log R_K$ when the three factors influencing SNR are increased independently. As any factor increases, estimation error remains high and even using as many as a million samples does not help much (see section 5.2 for setup details). Moreover, the steepness of the slopes in fig. 2 correlates inversely with the relative impact of the three factors. For instance, $\delta$ scales quadratically with the mismatch, and consequently, the corresponding error slope is the least steep, indicating this error is harder to reduce with more samples.

### 2.2. Analysis With Exact Inference And Conjugacy

This section considers theorem 2.1 for conjugate models. These models enjoy nice analytical properties, allowing us further insights into the low SNR problem. First, we discuss some conjugate model notation and then present the result.

Consider an exponential family

$$p(y|z) = h(y)\exp(T(y)^\top \phi(z) - A(z)),$$
$$A(z) = \log \int h(y)\exp(T(y)^\top \phi(z))dy, \qquad (8)$$

where $h(y)$ is the base measure, $T(y)$ is the sufficient statistic, $\phi$ is a one-to-one parameter map, and A is the log-partition function ensuring normalization. The corresponding conjugate family is

$$s(z|\xi) = \exp\left(\xi^\top \begin{bmatrix} \phi(z) \\ -A(z) \end{bmatrix} - B(\xi)\right),$$
$$B(\xi) = \log \int \exp\left(\xi^\top \begin{bmatrix} \phi(z) \\ -A(z) \end{bmatrix}\right)dz, \qquad (9)$$

where $\xi$ is the conjugate family parameter. $s$ is "conjugate" because if the prior is $p(z) = s(z|\xi_0)$ and the likelihood is $p(\mathcal{D}|z) = \prod_{y\in\mathcal{D}} p(y|z)$, then posterior is within the family and given by $p(z|\mathcal{D}) = s(z|\xi_\mathcal{D})$ where

$$\xi_\mathcal{D} = \xi_0 + \begin{bmatrix} \frac{\sum_{y\in\mathcal{D}} T(y)}{|\mathcal{D}|} \end{bmatrix}. \qquad (10)$$

**Corollary 2.3.** *Take a model with likelihood $p(\mathcal{D}|z)$ in an exponential family (eq. (8)) and prior $p(z) = s(z|\xi_0)$ in the conjugate family (eq. (9)). Let $R_K$ be as in eq. (2). Then, $SNR(R_K) = \sqrt{K}/\sqrt{\exp(\delta)^2 - 1}$ for*

$$\delta = \frac{1}{2}KL\left(s(z|\xi_{\mathcal{D}+\mathcal{D}^*}) \parallel s(z|\xi_\mathcal{D})\right)$$
$$+ \frac{1}{2}KL\left(s(z|\xi_{\mathcal{D}+\mathcal{D}^*}) \parallel s(z|\xi_{\mathcal{D}+2\mathcal{D}^*})\right) \qquad (11)$$
$$= \frac{B(\xi_\mathcal{D}) + B(\xi_{\mathcal{D}+2\mathcal{D}^*})}{2} - B(\xi_{\mathcal{D}+\mathcal{D}^*}), \qquad (12)$$

*where $\xi_\mathcal{D}$ is as in eq. (10) and B is as in eq. (9).*

This result is similar to theorem 2.1 (see section F for a standalone proof). The main advantage of this new result is

that the second form for $\delta$ in terms of log partition functions (eq. (12)) allows additional insight. Figure 5 plots the SNR for three conjugate models using eq. (12).

To understand eq. (12), note that $\xi_{\mathcal{D}+\mathcal{D}^*} = \frac{1}{2}(\xi_\mathcal{D} + \xi_{\mathcal{D}+2\mathcal{D}^*})$. Since $B$ is convex, $\delta$ is the *looseness in Jensen's inequality*: the mean of $B(\xi_\mathcal{D})$ and $B(\xi_{\mathcal{D}+2\mathcal{D}^*})$ versus $B$ applied to the mean of $\xi_\mathcal{D}$ and $\xi_{\mathcal{D}+2\mathcal{D}^*}$.

Jensen's inequality is tight when the function is nearly linear in the range evaluated. Imagine evaluating $B(a\xi)$ for $a > 0$, i.e., along a ray emanating from the origin. $B$ has a "log-sum-exp" form, so as $a$ becomes large, $B(a\xi)$ becomes nearly linear along this ray (Boyd & Vandenberghe, 2004). So, when $\xi_\mathcal{D}$ and $\xi_{\mathcal{D}+2\mathcal{D}^*}$ are large and lie along the ray, $\delta$ is small. In summary, $\delta$ is small (and the SNR large) when:

1. $\xi_\mathcal{D}$ is large (so that $B$ is locally "flat" near $\xi_\mathcal{D}$).

2. $\xi_\mathcal{D}$ and $\xi_{\mathcal{D}+2\mathcal{D}^*}$ lie close to a ray emanating from the origin. This happens when the statistics $T(\mathcal{D})$ and $T(\mathcal{D}^*)$ are similar, and the prior parameters $\xi_0$ are either small or nearly proportional to $\xi_\mathcal{D}$.

Overall, corollary 2.3 corroborates the results from theorem 2.1 and eq. (5). The log-partition function $B$ behaves like a "soft-max" function or, very informally, a "rounded cone." We know it is "rounded" near the origin, but if you follow any ray away from the origin, it becomes "flat." In these flat regions, Jensen's inequality is tight and SNR is high. Therefore, the impact of relative dataset size and mismatch of data statistics on SNR can be understood in terms of how they position points along the surface of the "cone" —small datasets position you near the origin where the log partition function is rounded, while different moments mean the points do not lie on a ray pointing near the origin. Either of these cases lead to a poor SNR. Below, we provide an example to visually demonstrate this phenomenon.

---

**Example: A Gaussian Conjugate Model**

Take the conjugate normal model with $p(z) = \mathcal{N}(z|0,1)$ and $p(y|z) = \mathcal{N}(y|z,1)$. Let $\overline{T}(\mathcal{D})$ denote the mean sufficient statistics of $y \in \mathcal{D}$. Take a training dataset $\mathcal{D}$ with $\overline{T}(\mathcal{D}) = 10$, a "matching" test dataset with $\overline{T}(\mathcal{D}_1^*) = 10$ and a "mismatched" test dataset with $\overline{T}(\mathcal{D}_2^*) = 5$. All datasets have 100 examples ($|\mathcal{D}| = |\mathcal{D}_1^*| = |\mathcal{D}_2^*| = 100$).

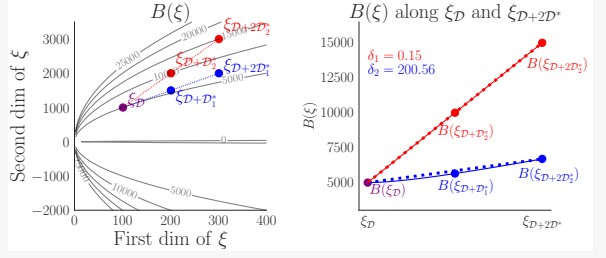

---

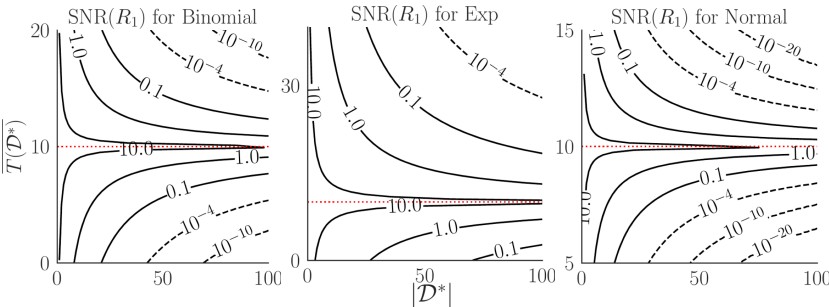

Figure 5: **SNR isocontours.** See section B for model details. For each model, we derive the log-normalization constant $B$ and use expressions from eq. (12) to calculate the SNR in closed form. We set the mean sufficient statistics $\overline{T}(\mathcal{D})$ to 10 (denoted in red dotted line) and the number of data points $|\mathcal{D}|$ to 1000. Setting these values theoretically is sufficient to calculate the SNR since eq. (12) requires the natural parameter $\xi$ under different datasets, which can be evaluated using eq. (10). Dataset mismatch increases as we move away from the red dotted line, and the relative size of $\mathcal{D}^*$ increases as we move along the horizontal axis. Either way, SNR decreases exponentially.

---

Figure 6: **Left**: Isocontours of the log partition function $B(\xi)$ (eq. (9)). **Right.** The values of $B(\xi)$ along the lines joining $\xi_{\mathcal{D}}$ to $\xi_{\mathcal{D}+\mathcal{D}_1^*}$ and $\xi_{\mathcal{D}+\mathcal{D}_2^*}$.

Figure 6 shows $B(\xi)$ along with the values of natural parameter $\xi$ for each dataset. Notice how $\xi_{\mathcal{D}}$, $\xi_{\mathcal{D}+\mathcal{D}_1^*}$, and $\xi_{\mathcal{D}+2\mathcal{D}_1^*}$ are equidistant on a "ray" pointing to near the origin (left panel), meaning Jensen's inequality is nearly tight (right panel), but the line joining $\xi_{\mathcal{D}}$ and $\xi_{\mathcal{D}+\mathcal{D}_2^*}$ does *not* point towards the origin, meaning Jensen's inequality is not tight, resulting in $d \approx 200.56$ and an astronomically small SNR of SNR $(R_1) \approx 7.9 \times 10^{-88}$.

## 3. Analysis With Approximate Inference

In the previous section, we assumed that inference was exact, i.e., $q_{\mathcal{D}}(z) = p(z|\mathcal{D})$. This allowed us to uncover the factors that affect the SNR. However, in practice, methods like VI or Laplace's approximation are not exact. This section generalizes our analysis to cases where $q_{\mathcal{D}}$ may not be the same as $p(z|\mathcal{D})$. We start by generalizing theorem 2.1 and then specialize it to conjugate models.

**Theorem 3.1.** *Let $R_K$ be as in eq. (2). Then, SNR $(R_K) = \sqrt{K}/\sqrt{\exp(\delta)^2 - 1}$ for*

$$\delta = \frac{1}{2}KL\left(q_{\mathcal{D}}(z|\mathcal{D}^*) \parallel q_{\mathcal{D}}(z)\right)$$

$$+ \frac{1}{2}KL\left(q_{\mathcal{D}}(z|\mathcal{D}^*) \parallel q_{\mathcal{D}}(z|2\mathcal{D}^*)\right) \qquad (13)$$

$$= \frac{1}{2}Z_{\mathcal{D}}(2\mathcal{D}^*) - Z_{\mathcal{D}}(\mathcal{D}^*), \qquad (14)$$

*where $Z_{\mathcal{D}}(\mathcal{D}^*) = \log \int p(\mathcal{D}^*|z)q_{\mathcal{D}}(z)dz$ and $q_{\mathcal{D}}(z|\mathcal{D}^*) \propto p(\mathcal{D}^*|z)q_{\mathcal{D}}(z)$.*

For a standalone proof, check section G. As before, eq. (13) determines $\delta$ in terms of divergences, but these distributions are unusual. One may think of $q_{\mathcal{D}}(z|\mathcal{D}^*)$ as the posterior

when using $q_{\mathcal{D}}(z)$ as a prior and conditioning on $\mathcal{D}^*$. When inference is exact, $q_{\mathcal{D}}(z|\mathcal{D}^*) = p(z|\mathcal{D} + \mathcal{D}^*)$ and eq. (13) reduces to eq. (3).

Similarly, eq. (14) is a generalization of eq. (4). To see this, write $\delta = 1/2\left(Z_{\mathcal{D}}(\emptyset) + Z_{\mathcal{D}}(2\mathcal{D}^*)\right) - Z_{\mathcal{D}}(\mathcal{D}^*)$ where $Z_{\mathcal{D}}(\emptyset) = \log \int q_{\mathcal{D}}(z)dz = 0$. When inference is exact, simple manipulations make the two expressions equal.

Next corollary specializes theorem 3.1 to the case of conjugate models. For simplicity, we assume that approximate distribution lies in the conjugate family.

**Corollary 3.2.** *Let $p(\mathcal{D}|z)$ and $p(z)$ be as in corollary 2.3. Let $q_{\mathcal{D}}(z) = s(z|\eta)$ be in the conjugate family (eq. (9)) with parameters $\eta$, and let $R_K$ be as in eq. (2). Then, $SNR(R_K) = \sqrt{K}/\sqrt{\exp(\delta)^2 - 1}$ for*

$$\delta = \frac{1}{2}KL\left(s(z|\eta + U(\mathcal{D}^*)) \parallel s(z|\eta)\right)$$

$$+ \frac{1}{2}KL\left(s(z|\eta + U(\mathcal{D}^*)) \parallel s(z|\eta + U(2\mathcal{D}^*))\right) \qquad (15)$$

$$= \frac{B(\eta) + B(\eta + U(2\mathcal{D}^*))}{2} - B(\eta + U(\mathcal{D}^*)), \qquad (16)$$

*where $B$ is as in eq. (9) and $U(\mathcal{D}) = \left[T(\mathcal{D}), |\mathcal{D}|\right]$.*

See section H for a proof. This result has the same functional form as corollary 2.3 and differs only in the canonical parameters. Now, $\eta$ are the parameters of $q_{\mathcal{D}}$ and $\eta + U(\mathcal{D}^*)$ are the parameters of the posterior obtained by conditioning on $\mathcal{D}^*$ with $q_{\mathcal{D}}$ as prior. When the inference is exact, $\eta = \xi_{\mathcal{D}}$, the above expressions reduce to corollary 2.3.

Note $\delta$ as in eq. (16) is again the looseness in Jensen's inequality: mean of $B(\eta)$ and $B(\eta + U(2\mathcal{D}^*))$ versus $B$ applied to mean of $\eta$ and $\eta + U(2\mathcal{D}^*)$. So, when test data statistics $U(\mathcal{D}^*)$ are such that $\eta$ and $\eta + U(2\mathcal{D}^*)$ lie along a ray originating at the origin, the Jensen's inequality in eq. (16) is tight and $\delta$ is small. Otherwise, we expect $\delta$ to be large and the SNR to be low.

Overall, theorem 3.1 and corollary 3.2 tell us that when approximate posterior closely resembles the true posterior, the relationships from section 2 hold as is. However, for arbitrary approximations, we can only reason about the SNR in terms of how much the "posteriors $q_{\mathcal{D}}(z|\mathcal{D}^*)$ and $q_{\mathcal{D}}(z|2\mathcal{D}^*)$" differ from the "prior $q_{\mathcal{D}}(z)$" as making more precise statements in terms of the datasets, like we did in section 2, will require specific assumptions on $q_{\mathcal{D}}$.

## 4. Learned Importance Sampling

The previous sections analyzed the SNR expressions for the simple MC estimator and found it decays quickly with increase in mismatch, dimensionality, and relative size of $\mathcal{D}^*$. When the SNR is poor, even with a large number of samples the estimation error remains high (fig. 2). This section considers an instance of the adaptive importance sampling idea (Owen, 2013, Chapter 10) as an illustrative way to mitigate the poor estimation error.

In general, when an MC estimator has high variance, a standard solution is to replace it with importance sampling (IS). For a valid proposal $r$, the IS estimator for PPD$_q$ can be written as

$$R_K^{IS} = \frac{1}{K} \sum_{k=1}^{K} \frac{p(\mathcal{D}^*|z_k)q_{\mathcal{D}}(z_k)}{r(z_k)} \quad \text{where} \quad z_k \sim r. \quad (17)$$

The choice of the proposal is delicate. Setting $r(z) = q_{\mathcal{D}}(z)$ reduces eq. (17) to the simple MC estimator, while using $r^{Opt} \propto p(\mathcal{D}^*|z)q(z|\mathcal{D})$—the optimal IS proposal—results in infinite SNR (Rainforth et al., 2020). The idea of adaptive IS is to somehow learn a good proposal $r$ (since $r^{Opt}$ is often intractable). Converting this idea into an algorithm is non-trivial as it requires making several decisions, like choosing the family of proposal distributions, an update scheme, and a stopping criterion (Owen, 2013, Algorithm 10.1).

One straightforward way to learn $r$, based on our analysis from sections 2 and 3, is to learn a parameterized proposal $r_w$, with parameters $w$, by optimizing the SNR of the estimator in eq. (17). However, maximizing SNR $(R_K^{IS})$ is equivalent to minimizing the variance of $R_K^{IS}$, which in turn is equivalent to minimizing the $\chi^2$-divergence between $r^{Opt}$ and $r_w$ (Dieng et al., 2017), and recent research suggests gradients for $\chi^2$ themselves suffer from poor SNR, making this direct optimization difficult (Geffner & Domke, 2021).

We take an alternative approach and learn $r_w$ by optimizing the importance weighted evidence lower-bound (IW-ELBO) (Burda et al., 2016). Let $z_m \sim r_w$. Then

$$\text{IW-ELBO}_M [r_w(z) \| p(\mathcal{D}^*|z)q_{\mathcal{D}}(z)]$$

$$:= \mathbb{E}\left[\log \frac{1}{M} \sum_{m=1}^{M} \frac{p(\mathcal{D}^*|z_m)q_{\mathcal{D}}(z_m)}{r_w(z_m)}\right]. \quad (18)$$

It is known that maximizing IW-ELBO in eq. (18) is *asymptotically equivalent* to minimizing the variance of $R^{IS}$,

or equivalently, maximizing SNR $(R^{IS})$ (Maddison et al., 2017; Dieng et al., 2017; Rainforth et al., 2018; Domke & Sheldon, 2018). More formally,

$$\lim_{M \to \infty} M\left(\log \text{PPD}_q - \text{IW-ELBO}_M\right)$$
$$= \mathbb{V}[R^{IS}]/(2\text{PPD}_q^2). \quad (19)$$

So, optimizing the IW-ELBO in eq. (18) can be thought of as a surrogate for optimizing the SNR of the IS estimator. The naive gradient estimator of IW-ELBO also has poor SNR (Rainforth et al., 2018; Finke & Thiery, 2019). Fortunately, a recently proposed re-parameterized gradient estimator circumvents this issue (Tucker et al., 2019; Finke & Thiery, 2019) and offers a practical option (Agrawal et al., 2020).

We refer to our specific instance of adaptive IS as the *Learned IS* (LIS). Figure 7 provides the pseudocode.

```
LearnedIS(D*, K)
    w ← Optimize(IW-ELBO)
    z_k ~ r_w   ∀k ∈ {1,...,K}
    R_K^IS ← 1/K Σ_{k=1}^{K} p(D*|z_k)q_D(z_k) / r_w(z_k)
```

Figure 7: Evaluating PPD$_q$ with Learned IS.

## 5. Examples

We consider four models: exponential family, linear regression, logistic regression, and a hierarchical model (see section B for more details of the setups). For each model, we use the simple MC estimator and demonstrate whenever either of the three factors is high, SNR is poor and PPD$_q$ estimates are biased even after using a large number of samples. We then use the learned IS estimator from fig. 7 and show that it enjoys significantly higher SNR and provides accurate estimates with fewer samples.

### 5.1. Exponential Family Models

Tables 1 and 2 show the results of PPD$_q$ estimation under exact inference and approximate inference, respectively. In both the cases, simple MC estimator suffers from low SNR and high estimation error even after using a million samples. The empirical SNR of $R_K^{IS}$ is much higher than $R_K$. Under exact inference, $R_K^{IS}$ is deterministically equal to PPD$_q$, and under approximate inference, the learned IS estimates are hundreds of nats higher than the simple MC estimates.

For the proposal and the variational families, we use full-rank Gaussians. For learned IS, we set $M = 16$ for the IW-ELBO and optimize for 1000 iterations with Adam (Kingma & Ba, 2015) and a learning rate of 0.001. (See section J for details, and table 6 for computation of $\log R$ and SNR $(R)$ in tables 1 and 2).

Table 1: Results for $\log \mathrm{PPD}_q$ estimation under *exact* inference (see table 5 for dataset details). We use $K = 10^6$ for simple MC and $K = 10^3$ for learned IS. Mean and standard deviation reported over ten runs. (See section B for model details.)

| Model | Ground truth $\log \mathrm{PPD}_q$ | Lower bound on $\log \mathrm{PPD}_q$ ($\uparrow$) | | Signal-to-noise ratio ($\uparrow$) | |
|---|---|---|---|---|---|
| | | Simple MC ($\mathbb{E}[\log R_K]$) | Learned IS ($\mathbb{E}[\log R_K^{\mathrm{IS}}]$) | Simple MC ($\mathrm{SNR}(R_K)$) | Learned IS ($\mathrm{SNR}(R_K^{\mathrm{IS}})$) |
| Normal | -774.64 | $-1183.98 \pm 0.34$ | $-774.64 \pm 0.00$ | $0.35 \pm 0.02$ | $76.5 \pm 23.43$ |
| Exp | -527.44 | $-559.98 \pm 0.16$ | $-527.44 \pm 0.00$ | $0.34 \pm 0.01$ | $222.76 \pm 147.59$ |
| Binomial | -327.42 | $-487.13 \pm 1.29$ | $-327.41 \pm 0.00$ | $0.03 \pm 0.00$ | $173.06 \pm 104.2$ |

Table 2: Results for $\log \mathrm{PPD}_q$ estimation under *approximate* inference (see table 5 for dataset details). We use $K = 10^6$ for simple MC and $K = 10^3$ for learned IS. Mean and standard deviation reported over ten runs.

| Model | Ground truth $\log \mathrm{PPD}_q$ | Lower bound on $\log \mathrm{PPD}_q$ ($\uparrow$) | | Signal-to-noise ratio ($\uparrow$) | |
|---|---|---|---|---|---|
| | | Simple MC ($\mathbb{E}[\log R_K]$) | Learned IS ($\mathbb{E}[\log R_K^{\mathrm{IS}}]$) | Simple MC ($\mathrm{SNR}(R_K)$) | Learned IS ($\mathrm{SNR}(R_K^{\mathrm{IS}})$) |
| Normal | – | $-1194.32 \pm 0.41$ | $-775.23 \pm 0.00$ | $0.35 \pm 0.02$ | $238.79 \pm 172.46$ |
| Exp | – | $-576.27 \pm 0.14$ | $-542.34 \pm 0.00$ | $0.36 \pm 0.01$ | $215.09 \pm 140.52$ |
| Binomial | – | $-382.46 \pm 0.74$ | $-322.66 \pm 0.00$ | $0.34 \pm 0.02$ | $70.13 \pm 35.29$ |

Table 3: Results for $\log \mathrm{PPD}_q$ estimation for logistic regression (section 5.3). We use $K = 10^6$ for simple MC and $K = 10^3$ for IS estimators. Mean and standard deviation reported over ten runs.

| Model | Ground truth $\log \mathrm{PPD}_q$ | Lower bound on $\log \mathrm{PPD}_q$ ($\uparrow$) | | Signal-to-noise ratio ($\uparrow$) | |
|---|---|---|---|---|---|
| | | Simple MC ($\mathbb{E}[\log R_K]$) | Learned IS ($\mathbb{E}[\log R_K^{\mathrm{IS}}]$) | Simple MC ($\mathrm{SNR}(R_K)$) | Learned IS ($\mathrm{SNR}(R_K^{\mathrm{IS}})$) |
| Baseline | - | $-525.25 \pm 0.01$ | $-525.12 \pm 0.00$ | $1.35 \pm 0.20$ | $645.67 \pm 14.99$ |
| More dimension | - | $-702.07 \pm 0.28$ | $-543.00 \pm 0.00$ | $0.04 \pm 0.01$ | $57.78 \pm 1.37$ |
| More mismatch | - | $-1687.98 \pm 0.96$ | $-734.32 \pm 0.00$ | $0.03 \pm 0.00$ | $728.63 \pm 16.01$ |
| More test data | - | $-5143.69 \pm 0.34$ | $-5097.60 \pm 0.00$ | $0.04 \pm 0.01$ | $802.34 \pm 18.37$ |

Table 4: Results for $\log \mathrm{PPD}_q$ estimation for MovieLens 25M dataset (section 5.4). Mean and standard deviation reported over ten runs.

| Method | No. of samples ($K$) | Lower bound on $\log \mathrm{PPD}_q$ ($\uparrow$) | | Signal-to-noise ratio ($\uparrow$) | |
|---|---|---|---|---|---|
| | | Simple MC ($\mathbb{E}[\log R_K]$) | Learned IS ($\mathbb{E}[\log R_K^{\mathrm{IS}}]$) | Simple MC ($\mathrm{SNR}(R_K)$) | Learned IS ($\mathrm{SNR}(R_K^{\mathrm{IS}})$) |
| Flow VI | $K = 10^3$ | $-796.24 \pm 0.13$ | $-779.39 \pm 0.02$ | $0.05 \pm 0.02$ | $0.11 \pm 0.04$ |
| | $K = 10^6$ | $-787.27 \pm 0.08$ | $-777.73 \pm 0.01$ | $0.04 \pm 0.01$ | $0.48 \pm 0.29$ |
| Gaussian VI | $K = 10^3$ | $-828.22 \pm 0.17$ | $-783.89 \pm 0.03$ | $0.04 \pm 0.00$ | $0.12 \pm 0.01$ |
| | $K = 10^6$ | $-811.61 \pm 0.13$ | $-781.88 \pm 0.02$ | $0.04 \pm 0.01$ | $0.32 \pm 0.13$ |

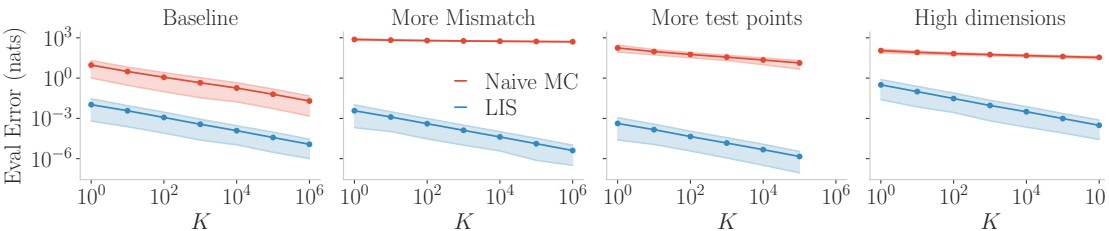

Figure 8: **Estimation Error.** We plot the estimation error, $\log \text{PPD}_q - \log R_K$, against the number of samples $K$ for the linear regression model (section 5.2). The fifth and the ninety-fifth percentiles are represented as the filled regions. Across the scenarios, the estimation error worsens as either of the three factors increase. LIS significantly reduces the error. See section K for more details on scenarios.

## 5.2. Linear Regression

We take the linear regression model with exact inference as described in theorem 2.2 and construct four scenarios. We start with a baseline where none of the three factors influencing SNR are too high. We then independently increase mismatch, dimensionality, and relative size (see section K).

In fig. 8, we plot the error in evaluating $\log \text{PPD}_q$ under exact inference using simple MC and learned IS. For the baseline (first panel), the simple MC has high enough SNR, and the estimation is accurate for $K = 10^6$. This error increases dramatically when either of the factors increase (see red curves in last three panels). The learned IS consistently evaluates accurately across all scenarios (see blue curves). (See section K for details of learned IS settings).

## 5.3. Logistic Regression

We construct four scenarios similar to section 5.2—the baseline where none of the three factors are too high and additional scenarios where each factor is increased individually.

Table 3 reports the results from estimating $\text{PPD}_q$ using the simple MC estimator and the learned IS estimator. For baseline, the simple MC estimator has reasonably high SNR but for other scenarios it suffers from low SNR and poor estimation error. The learned IS consistently provides higher SNRs and better estimates. See section L for setup details.

## 5.4. Hierarchical Model

We consider a hierarchical user-preference model for the MovieLens 25M dataset. Here, we have no dataset shift and the relative size is small ($|\mathcal{D}^*|/|\mathcal{D}| = 0.1$). However, the dimensionality is moderately high ($d = 1065$) and simple MC estimator can still suffer from low SNR. We consider two posterior variational approximations—full-rank Gaussians and normalizing flows—for learning $q_{\mathcal{D}}$.

Table 4 reports $\log \text{PPD}_q$ estimates using simple MC and learned IS for different values of $K$. When using the simple MC with $K = 10^6$, flow VI reports $\text{PPD}_q$ more than 20 nats higher than Gaussian VI (second column). However, the

SNR of these estimates is extremely low. With learned IS, flow VI is only 4 nats higher than the Gaussian VI, with much higher SNR (fourth column). So, while flow VI may be better, the difference is not as large as it seems initially. (See section M for setup details.)

## 6. Discussion

**Related Work.** Wu et al. (2017) explored the use of Annealed Importance Sampling (AIS) (Neal, 2001) for estimating the posterior predictive density in decoder based models. In particular, they used AIS for estimating the normalization constant of the unnormalized density $p(y_i^*|z_i)q(z_i|\mathcal{D})$ for each data point $y_i$ in the test data set $\mathcal{D}^*$. Different from them, we focus on black-box treatment of probabilistic models (Ranganath et al., 2014; Kucukelbir et al., 2017) and exploit BBVI schemes (Kucukelbir et al., 2017; Agrawal et al., 2020) for estimating the posterior predictive densities over $\mathcal{D}^*$. Recent theoretical advances (Domke, 2019; 2020; Domke et al., 2024; Kim et al., 2023; 2024b;a) make BBVI a general purpose inference method that is reliably applicable to a wide range of problems (Carpenter et al., 2017).

Other research has explored learning approximate posterior distributions $q_{\mathcal{D}}$ to calibrate for test-time utilities (Stoyanov et al., 2011; Lopez et al., 2020; Lacoste–Julien et al., 2011; Morais & Pillow, 2022; Kuśmierczyk et al., 2019; Knoblauch et al., 2022; Kuśmierczyk et al., 2020; Vadera et al., 2021). Such methods aim to learn a distribution $q'$ that is different from $q_{\mathcal{D}}$ and optimizes the expectation of some utility function under $q_{\mathcal{D}}$ at test-time. We focus on the problem of estimating the posterior predictive density for a given $q_{\mathcal{D}}$ at test-time, and do not change the given posterior; we simply focus on accurate estimation.

Ruiz et al. (2016) explored learning an importance sampling estimator for estimating the gradients for BBVI (Ranganath et al., 2014). They learn a proposal distribution $r$ while learning the parameters of the variational distribution $q_{\mathcal{D}}$, and rely on exponential families for closed-form updates. We do not focus on learning the variational distribution $q_{\mathcal{D}}$, and use BBVI methods for learning the proposal that

can be in any suitable family of distributions (Rezende & Mohamed, 2015; Papamakarios et al., 2021; Webb et al., 2019; Agrawal et al., 2020).

Vehtari et al. (2016) evaluate predictive accuracy using metrics that involve leave-one-out "point wise" predictive density of the type $p(y_i|\mathcal{D}_{-i})$ over the training data $\mathcal{D}$. To estimate $p(y_i|\mathcal{D}_{-i}) = \int p(y_i|z)p(z|\mathcal{D}_{-i})dz$, the authors consider using the full posterior distribution $p(z|\mathcal{D})$ as the proposal. However, $p(z|\mathcal{D})$ can have thinner tails than $p(z|\mathcal{D}_{-i})$ leading to large importance weights. To remedy this, the authors fit a Pareto distribution to the importance weights and then use statistics from the fitted distribution for final estimation. While the PSIS-LOO setting differs from our focus, one can use PSIS ideas to improve LIS estimates if $r$ is suspected of thin tails.

Rainforth et al. (2020) propose a framework for target-aware Bayesian inference (TABI) in which they decompose the posterior expectations into three components. Each of the three components is then computed as an instance of importance sampling using the Annealed Importance Sampling (AIS) or Nested Importance Sampling (NIS). One can apply the TABI framework for $\text{PPD}_q$ estimation; however, after some simple observations, this reduces to estimating $\int p(\mathcal{D}|z)q_{\mathcal{D}}(z)dz$ with AIS or NIS (and is same as the approach from Wu et al. (2017)). In recent work, Llorente et al. (2023) extend the TABI framework by employing the generalized thermodynamic integration scheme (GIS) for solving the posterior expectations. When placing these TABI approaches in context, it is crucial to note that we focus on approximate inference problems. Running MCMC procedures like AIS or thermodynamic integrations procedures like GIS is often infeasible or extremely slow on such problems (due to a large number of data points or dimensions), and view the MCMC procedures as orthogonal to our variational approach.

Reichelt et al. (2022a) propose the concept of expectation programming, where a probabilistic programming system considers the target posterior expectation as a first-class citizen. They aim to build an efficient estimation pipeline when target functions are previously known. In their implementation, they currently use Annealed Importance Sampling as the choice of inference scheme. Our proposed methodology can join their suite of inference options when the target functions are more amenable to a variational formulation.

Izmailov et al. (2021a) point out that the posteriors in Bayesian neural network can be bad at generalizing under specific dataset shifts. They uncover pathologies in the BNN posteriors that lead to poor generalization and present techniques that can possibly mitigate these. Different from them, we focus on understanding the problem of inaccurate $\text{PPD}_q$ estimation and how to improve estimation without changing the properties of the posterior.

Kristiadi et al. (2022) hypothesize that the inaccuracy of the posterior predictive estimation is related to the inaccuracy of the approximate posterior. The authors explore this empirically and consider the use of normalizing flows to get better posterior approximations. We find this work orthogonal to ours. We lay out the conditions that lead to poor posterior predictive estimations (including when inference is exact) and provide a post hoc method for improving the estimation without changing the approximate posterior.

**Conclusions.** This paper develops a theoretical understanding of why the SNR of the naive PPD estimator can be extremely poor. We demonstrate that even if the inference is exact, SNR suffers when either of the following quantities increases: mismatch between the training and test data, dimensionality of the latent space, or the size of the test data relative to training data. We hope our work serves as a strong cautionary note and more practitioners monitor (and report) the SNR value of their PPD estimates. Moreover, whenever the SNR values of the naive estimator is low, our work shows that this does not necessarily reflect on the accuracy of the approximate posterior but simply on our ability to accurately estimate PPD. In cases of low SNR, we suggest practitioners use approaches like LIS to improve the reliability of the estimates.

**Limitations.** The primary contribution of this work is theoretical. Our explorative strategy performs favorably and can be developed further in the future work. The proposal learning can be computationally expensive when the test data is large. Further, learned IS requires evaluating the approximate distribution and does not immediately extend to Markov chain Monte Carlo methods.

# Acknowledgement

The authors would like to thank Edmond Cunnigham, Mohit Yadav, and Miguel Fuentes for useful discussions and comments, and Michelle Hedlund and Ankit Patel for helping improve the clarity of the manuscript.

# Impact Statement

This paper presents work whose goal is to advance the field of Machine Learning. There are many potential societal consequences of our work, none which we feel must be specifically highlighted here.

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

# A. Additional Comments

Here we offer some additional comments on the different aspects of our formulation that we could not cover in the main text.

## A.1. True posterior predictive distribution vs. $\text{PPD}_q$

In this paper, we are not interested in estimating the posterior predictive distribution under the true posterior ($\int p(\mathcal{D}^*|z)p(z|\mathcal{D})dz$). We are interested in estimating the posterior predictive distribution under the approximate posterior as defined in eq. (1). Accurate measurements of the $\text{PPD}_q$ are of independent interest and are used for forecasting, predictions, and inference method comparisons (see citations in discussion after eq. (1)).

## A.2. Relationship of estimation error and variance of $R_K$

Let $G$ be an unbiased estimator of the $\text{PPD}_q$ such that $\mathbb{E}[G] = \text{PPD}_q$. Then, using Jensen's inequality, we know that $\mathbb{E}[\log G] \leq \log \text{PPD}_q$, making $\log G$ a biased estimator of $\log \text{PPD}_q$. This bias or the looseness in Jensen's inequality is directly related to the variance or inversely related to the SNR of $G$. To see this, consider the $K$ sample estimator $G_K$. A second-order Taylor series approximation in the vicinity of $\log \text{PPD}_q$ (as done in the second-order delta method) gives the following about the bias or the looseness in Jensen's inequality:

$$\log \text{PPD}_q - \mathbb{E}[\log G_K] \approx \frac{1}{2K} \frac{V[G]}{\text{PPD}_q^2} = \frac{1}{2K} \frac{1}{\text{SNR}(G)^2}.$$

These relationships hold when $G$ is unbiased and apply to the estimators in eqs. (2) and (17). Therefore, we want the SNR to be high for lower estimation errors.

## A.3. SNR vs variance of $R_K$

In this paper, we use signal-to-noise ratio (SNR) of $R_K$ defined as $\text{SNR}(R_K) = \mathbb{E}[R_K]/\sqrt{\mathbb{V}[R_K]}$. We study SNR because it is equivalent to relative variance and bakes in the idea of how large the estimator variance is relative to the target quantity. The idea of relative variance becomes crucial when the target quantity is numerically small, as in the case of $\text{PPD}_q$ values. To make this precise, let's consider an example where the $\log \text{PPD}_q = -100$ (for reference, all estimates of $\log \text{PPD}_q$ in tables 1 to 4 are lower than $-100$). Also, consider an unbiased estimator $R$ with variance $\mathbb{V}(R) = 10^{-20}$. In the absolute sense, the variance of this estimator is low; however, $R$ carries more noise than signal. To see why, note that $\text{PPD}_q = \exp(-100) \approx 3.72 \times 10^{-44}$. Intuitively, $R$ varies on the scale of $10^{-10}$ (standard deviation) and will produce noisy approximations of the target value that is the order of $10^{-44}$. SNR naturally captures this intuition: $\text{SNR}(R) = \text{PPD}_q/\sqrt{\mathbb{V}(R)} \approx 3.72 \times 10^{-34}$ and flags the estimator as poor.

## A.4. Joint vs. Marginal PPD

The usage of "joint" PPD (as in eq. (1)) depends on both—the model and the modeler's preference. The joint PPD ($\int p(\mathcal{D}^*|z)q_{\mathcal{D}}(z)dz$) is a natural metric whenever the test data are not IID (Williams et al., 2020) or one wants to measure correlations between new data points (Wang et al., 2021). It also shows up in the conditional log marginal likelihood metric, where a fraction of observations is used to evaluate the conditional likelihood over the remaining observations (Lotfi et al., 2022). The "margina" PPD ($\int p(y^*|z)q_{\mathcal{D}}(z)dz$) is a natural metric when one wants to measure the model's ability to predict a single data point. Moreover, the dependence of SNR on the size of the test data is relatively weaker than the dependence on the mismatch and dimensionality (eqs. (6) and (7)).

# B. Model Details

This section provides high-level overview of the models considered throughout the paper. For the first three models, we use synthetic data sampled from the model. For the hierarchical model, we use the MovieLens 25M dataset. For more details of the setup, see sections J to M.

**Exponential Family Models.** We consider three exponential family models. A binomial model where $p(y|z)$ is a binomial distribution with a known number of trials $n$ and unknown success probability $z \in [0, 1]$, and $p(z)$ is a beta distribution. An exponential model where $p(y|z)$ is an exponential distribution with the unknown rate $z \in \mathbb{R}_+$, and $p(z)$ is a gamma

distribution. A normal model where $p(y|z)$ is a normal distribution with known variance $\sigma^2$ and unknown mean $z \in \mathbb{R}$, and $p(z)$ is also a normal distribution. See table 9 in section J for full model details.

Table 5: Summary of the data sets used for results in tables 1 and 2.

| Model | $\overline{T(\mathcal{D})}$ | $|\mathcal{D}|$ | $\overline{T(\mathcal{D}^*)}$ | $|\mathcal{D}^*|$ | $\delta$ |
|---|---|---|---|---|---|
| Normal | 10.08 | 100 | 4.96 | 100 | 210.85 |
| Exp | 7.00 | 100 | 39.37 | 100 | 11.74 |
| Binomial | 8.96 | 100 | 41.06 | 100 | 23.32 |

**Linear Regression.**   We set $p(y|z) = \mathcal{N}(y|z^\top x, 1)$ and $p(z) = \mathcal{N}(z|0, I)$, where $x \in \mathbb{R}^d$ and $y \in \mathbb{R}$ in model from section 2.1. We use this model for the example in figs. 2 and 4 and experiments in section 5.2. For more details on the scenarios in figs. 2 and 8, see section K

**Logistic Regression.**   The non-conjugate model with $p(y|z) = \mathcal{B}(y|\text{sigmoid}(z^\top x))$ and $p(z) = \mathcal{N}(z|0, I)$ where response $y \in \{0, 1\}$, latent variable $z \in \mathbb{R}^d$, and feature vector $x \in \mathbb{R}^d$. We use this model in section 5.3 experiments. For approximate inference, we use a full-rank Gaussian variational posterior. For more details, see section L

**Hierarchical Model.**   The hierarchical model using the MovieLens 25M—a dataset of 25 million movie ratings along with a set of features for each movie (Vig et al., 2012). We randomly select 100 users after filtering those with more than 1000 ratings. We keep one-tenth of the ratings for test data, and PCA the features to ten dimensions (see section M for more details).

The task is to model rating $y_{i,j} \in \{0, 1\}$ of user $i$ for movie $j$ with given features $x_{i,j}$. We use the model

$$p(\theta, \lambda, y|x) = \mathcal{N}(\theta|0, I) \times \prod_{i=1}^{100} \mathcal{N}(\lambda_i|\mu(\theta), \Sigma(\theta)) \prod_{i=1}^{n_i} \mathcal{B}(y_{i,j}|\text{sigmoid}(\lambda_i^\top x_{i,j})) \tag{20}$$

where $\theta$ and $\lambda$ together represent latent variables $z$; $\theta$ are the global latent variables capturing preferences over users and $\lambda_i$ are the local latent variables capturing preferences for user $i$. $\mu$ and $\Sigma$ are functions such that if $\theta = [\theta_\mu, \theta_\Sigma]$, $\mu(\theta) = \theta_\mu$ and $\Sigma(\theta) = \text{tril}(\theta_\Sigma)^\top \text{tril}(\theta_\Sigma)$, where tril takes an unconstrained vector and outputs a lower-triangular positive definite matrix. $n_i$ is the number of ratings for user $i$. $\mathcal{B}$ is the Bernoulli distribution. We use this model for illustration in fig. 1 and experiments in Section 5.4.

## C. Proof for Theorem 2.1

For the proof sketch, see the theorem 2.1 in the main text. Here, we first provide the lemmas and corollaries needed for the proof and then provide the proof itself. Lemma C.1 rewrites the SNR of the naive Monte Carlo estimator from eq. (2) as a function of moments of the estimators. Lemma C.3 then provides the general expression for the moments of the naive Monte Carlo estimator. Combining lemmas C.1 and C.3, gives the SNR in the form normalizing constants as in eq. (4). Lemma C.4 gives the KL divergence between any two posterior distributions. Lemma C.5 then uses the result in lemma C.4 to give the expression for the sum of the KL divergences between three posterior distributions, where the first distribution is the same between the two KL terms (this is the general structure that appears in the eq. (3)). Corollary C.6 simplifies this sum KL expression for datasets with a particular structure. Finally, we prove the theorem 2.1 by combining all the above results.

**Lemma C.1.** *Let $R_K$ be the Monte Carlo estimator in eq. (2). Then,*

$$SNR\left(R_K\right) = \frac{\sqrt{K}}{\sqrt{\exp\left(\delta\right)^2 - 1}}, \quad where \ \delta = \frac{1}{2}\log\left(\frac{\mathbb{E}[R_1^2]}{\mathbb{E}[R_1]^2}\right) \tag{21}$$

*Proof.* The proof follows naturally from the definition of SNR $\left(R_K\right)$.

$$\text{SNR}\left(R_K\right) = \sqrt{K}\text{SNR}\left(R_1\right) = \sqrt{K}\frac{\mathbb{E}[R_1]}{\sqrt{\mathbb{V}[R_1]}} \tag{22}$$

$$= \sqrt{K}\frac{\mathbb{E}[R_1]}{\sqrt{\mathbb{E}[R_1^2] - \mathbb{E}[R_1]^2}} \tag{23}$$

$$\overset{(a)}{=} \frac{\sqrt{K}}{\sqrt{\left(\frac{\mathbb{E}[R_1^2]}{\mathbb{E}[R_1]^2} - 1\right)}} \tag{24}$$

$$\overset{(b)}{=} \frac{\sqrt{K}}{\sqrt{\exp\left(2\delta\right) - 1}} = \frac{\sqrt{K}}{\sqrt{\exp\left(\delta\right)^2 - 1}}, \tag{25}$$

where (a) follows from the fact LHS and RHS of $\overset{(a)}{=}$ are equal for $\mathbb{E}[R_1] > 0$ and limit is the same at $\mathbb{E}[R_1] = 0$; and (b) follows from the definition of $\delta$. $\qquad\square$

**Definition C.2** (Log-normalization function). Let $\mathcal{D}$ be some dataset. Let $p(\mathcal{D}|z)$ be the likelihood and $p(z)$ be the prior. Then, posterior distribution $p(z|\mathcal{D}) = \frac{p(\mathcal{D}|z)p(z)}{\exp V(\mathcal{D})}$, where

$$V(\mathcal{D}) := \log \int p(\mathcal{D}|z)p(z)dz. \tag{26}$$

**Lemma C.3.** *Let $p(\mathcal{D}|z)$ be the likelihood and $p(z)$ be the prior. Let $\mathcal{D}^*$ be some test data. Let $p(\mathcal{D} + \mathcal{D}^*|z) = p(\mathcal{D}|z)p(\mathcal{D}^*|z)$ for any $\mathcal{D}$ and $\mathcal{D}^*$. Let $R_1$ be the Monte Carlo estimator for the $PPD_q$ under exact inference (eq. (2) with $K = 1$ and $q_{\mathcal{D}}(z) = p(z|\mathcal{D})$). Then,*

$$\mathbb{E}\left[R_1^c\right] = \frac{\exp V\left(\mathcal{D} + c\mathcal{D}^*\right)}{\exp V\left(\mathcal{D}\right)}, \tag{27}$$

*where $c$ is a non-negative integer and $V$ is as in definition C.2.*

*Proof.* The proof is straightforward for $c = 0$. For $c \geq 1$, we have

$$\mathbb{E}\left[R_1^c\right] \stackrel{(a)}{=} \mathbb{E}\left[p(\mathcal{D}^*|z)^c\right] \stackrel{(b)}{=} \mathbb{E}\left[p(c\mathcal{D}^*|z)\right] \tag{28}$$

$$= \int p(c\mathcal{D}^*|z)p(z|\mathcal{D})dz \tag{29}$$

$$= \frac{\int p(c\mathcal{D}^*|z)p(\mathcal{D}|z)p(z)dz}{\exp V(\mathcal{D})} \tag{30}$$

$$\stackrel{(c)}{=} \frac{\int p(\mathcal{D} + c\mathcal{D}^*|z)p(z)dz}{\exp V(\mathcal{D})} \tag{31}$$

$$\stackrel{(d)}{=} \frac{\exp V(\mathcal{D} + c\mathcal{D}^*)}{\exp V(\mathcal{D})}. \tag{32}$$

where $(a)$ follows from definition of eq. (2), $(b)$ and $(c)$ follow from the i.i.d assumption on the datasets, and $(d)$ follows from definition C.2. Note: we do not require points within a dataset to be i.i.d. $\qquad \square$

**Lemma C.4.** *Let $p(\mathcal{D}|z)$, $p(z)$, and $p(z|\mathcal{D})$ be as in definition C.2. Let $\mathcal{D}_a$ and $\mathcal{D}_b$ be the two multisets of data. Then,*

$$KL\left(p(z|\mathcal{D}_a) \,\|\, p(z|\mathcal{D}_b)\right) \tag{33}$$

$$= \mathbb{E}\left[\log \frac{p(\mathcal{D}_a|z)}{p(\mathcal{D}_b|z)}\right] - V(\mathcal{D}_a) + V(\mathcal{D}_b) \tag{34}$$

$$\tag{35}$$

*Proof.*

$$KL\left(p(z|\mathcal{D}_a) \,\|\, p(z|\mathcal{D}_b)\right) \tag{36}$$

$$= \mathbb{E}\left[\log \frac{p(z|\mathcal{D}_a)}{p(z|\mathcal{D}_b)}\right] \tag{37}$$

$$= \mathbb{E}\left[\log \frac{\frac{p(\mathcal{D}_a|z)p(z)}{\exp(V(\mathcal{D}_a))}}{\frac{p(\mathcal{D}_b|z)p(z)}{\exp V(\mathcal{D}_b)}}\right] \tag{38}$$

$$= \mathbb{E}\left[\log \frac{p(\mathcal{D}_a|z)}{p(\mathcal{D}_b|z)}\right] - V(\mathcal{D}_a) + V(\mathcal{D}_b) \tag{39}$$

$$\tag{40}$$

$$\square$$

**Lemma C.5.** *Let $p(\mathcal{D}|z)$, $p(z)$, and $p(z|\mathcal{D})$ be as in definition C.2. Let $\mathcal{D}_1$, $\mathcal{D}_2$, and $\mathcal{D}_3$ be the three multisets of data. Then,*

$$\frac{1}{2}KL\left(p(z|\mathcal{D}_3) \,\|\, p(z|\mathcal{D}_1)\right) + \frac{1}{2}KL\left(p(z|\mathcal{D}_3) \,\|\, p(z|\mathcal{D}_2)\right) \tag{41}$$

$$= \mathbb{E}\left[\log p(\mathcal{D}_3|z) - \frac{1}{2}\log p(\mathcal{D}_1|z) - \frac{1}{2}\log p(\mathcal{D}_2|z)\right] \tag{42}$$

$$+ \frac{V(\mathcal{D}_1) + V(\mathcal{D}_2)}{2} - V(\mathcal{D}_3). \tag{43}$$

*Proof.* Applying lemma C.4 to $\mathcal{D}_3$ and $\mathcal{D}_1$ gives

$$KL\left(p(z|\mathcal{D}_3) \,\|\, p(z|\mathcal{D}_1)\right) \tag{44}$$

$$= \mathbb{E}\left[\log p(\mathcal{D}_3|z) - \log p(\mathcal{D}_1|z)\right] - V(\mathcal{D}_3) + V(\mathcal{D}_1) \tag{45}$$

$$\tag{46}$$

and applying it to $\mathcal{D}_3$ and $\mathcal{D}_2$ gives

$$\text{KL}\left(p(z|\mathcal{D}_3) \| p(z|\mathcal{D}_2)\right) \tag{47}$$

$$= \mathbb{E}\left[\log p(\mathcal{D}_3|z) - \log p(\mathcal{D}_2|z)\right] - V(\mathcal{D}_3) + V(\mathcal{D}_2). \tag{48}$$

$$\tag{49}$$

Now, multiplying the above two equations by $\frac{1}{2}$ and adding them gives

$$\frac{1}{2}\text{KL}\left(p(z|\mathcal{D}_3) \| p(z|\mathcal{D}_1)\right) + \frac{1}{2}\text{KL}\left(p(z|\mathcal{D}_3) \| p(z|\mathcal{D}_2)\right) \tag{50}$$

$$= \frac{1}{2}\mathbb{E}\left[\log p(\mathcal{D}_3|z) - \log p(\mathcal{D}_1|z)\right] - \frac{1}{2}V(\mathcal{D}_3) + \frac{1}{2}V(\mathcal{D}_1) \tag{51}$$

$$+ \frac{1}{2}\mathbb{E}\left[\log p(\mathcal{D}_3|z) - \log p(\mathcal{D}_2|z)\right] - \frac{1}{2}V(\mathcal{D}_3) + \frac{1}{2}V(\mathcal{D}_2) \tag{52}$$

$$= \mathbb{E}\left[\log p(\mathcal{D}_3|z) - \frac{1}{2}\log p(\mathcal{D}_1|z) - \frac{1}{2}\log p(\mathcal{D}_2|z)\right] \tag{53}$$

$$+ \frac{V(\mathcal{D}_1) + V(\mathcal{D}_2)}{2} - V(\mathcal{D}_3). \tag{54}$$

$$\square$$

**Corollary C.6.** *Let $p(\mathcal{D}|z)$, $p(z)$, and $p(z|\mathcal{D})$ be as in definition C.2. Let $\mathcal{D}_1 = c_a\mathcal{D}$, $\mathcal{D}_2 = c_a\mathcal{D} + 2c_b\mathcal{D}^*$, and $\mathcal{D}_3 = c_a\mathcal{D} + c_b\mathcal{D}^*$ be the three multisets of data where $c_a$ and $c_b$ are non-negative integers. Then,*

$$\frac{1}{2}KL\left(p(z|\mathcal{D}_3) \| p(z|\mathcal{D}_1)\right) + \frac{1}{2}KL\left(p(z|\mathcal{D}_3) \| p(z|\mathcal{D}_2)\right) \tag{55}$$

$$= \frac{V(\mathcal{D}_1) + V(\mathcal{D}_2)}{2} - V(\mathcal{D}_3). \tag{56}$$

**Theorem C.7** (Repeated for convenience). *Let $p(\mathcal{D}|z)$ be the likelihood and $p(z)$ be the prior. Let $\mathcal{D}^*$ be some test data. Let $p(\mathcal{D} + \mathcal{D}^*|z) = p(\mathcal{D}|z)p(\mathcal{D}^*|z)$ for any $\mathcal{D}$ and $\mathcal{D}^*$. Let $R_K$ (as in eq. (2)) be the Monte Carlo estimator for the $PPD_q$ under exact inference. Then, the signal-to-noise ratio of $R_K$ is given by $SNR(R_K) = \sqrt{K}/\sqrt{\exp(\delta)^2 - 1}$ where*

$$\delta = \frac{1}{2}KL\left(p(z|\mathcal{D} + \mathcal{D}^*) \| p(z|\mathcal{D})\right) + KL\left(p(z|\mathcal{D} + \mathcal{D}^*) \| p(z|\mathcal{D} + 2\mathcal{D}^*)\right) \tag{57}$$

$$= \frac{V(\mathcal{D}) + V(\mathcal{D} + 2\mathcal{D}^*)}{2} - V(\mathcal{D} + \mathcal{D}^*) \tag{58}$$

*where $V$ is as in definition C.2.*

*Proof.*

$$\delta \overset{(a)}{=} \frac{1}{2}\log\frac{\mathbb{E}[R_1^2]}{\mathbb{E}[R_1]^2} \tag{59}$$

$$= \frac{1}{2}\log\mathbb{E}\left[R_1^2\right] - \log\mathbb{E}\left[R_1\right] \tag{60}$$

$$\overset{(b)}{=} \frac{1}{2}\log\frac{\exp V(\mathcal{D} + 2\mathcal{D}^*)}{\exp V(\mathcal{D})} - \log\frac{\exp V(\mathcal{D} + \mathcal{D}^*)}{\exp V(\mathcal{D})} \tag{61}$$

$$\overset{(c)}{=} \frac{V(\mathcal{D} + 2\mathcal{D}^*) + V(\mathcal{D})}{2} - V(\mathcal{D} + \mathcal{D}^*) \tag{62}$$

(a) follows from lemma C.1, (b) follows from lemma C.3, and (c) follows from some simple algebraic manipulations. Now, for the KL-divergence result, if we take the expression in corollary C.6, and plug $\mathcal{D}_1 = \mathcal{D}$ and $\mathcal{D}_2 = \mathcal{D} + 2\mathcal{D}^*$ and $\mathcal{D}_3 = \mathcal{D} + \mathcal{D}^*$, then we get the same expression as eq. (58). $\quad\square$

# D. Discussion for informal result in eq. (5)

In this section, we provide a detailed discussion for the informal result in eq. (5). First, we provide a lemma and a corollary that will be used in arriving at the main result: Lemma D.1 gives the KL divergence between any two Gaussians and Corollary D.2 simplifies the expression for the sum of the KL divergences between three Gaussians, where the first distribution is the same between the two KL terms (this structure appears in the SNR expressions in eq. (3)). Finally, we restate the assumptions and calculate the result in eq. (5) using these results.

At the end of the section, we provide a note on the convergence of KL divergence and why it requires additional assumptions than just the convergence of the distributions.

**Lemma D.1.** *Let $\mathcal{N}(\mu_0, \Sigma_0)$ and $\mathcal{N}(\mu_1, \Sigma_1)$ be two Gaussian distributions of dimensionality $d$ with $\Sigma_0, \Sigma_1 \succ 0$. Then,*

$$
\begin{aligned}
KL\left(\mathcal{N}(\mu_0, \Sigma_0) \,\|\, \mathcal{N}(\mu_1, \Sigma_1)\right) = \operatorname{tr}&\left(\frac{1}{2}\Sigma_1^{-1}\Sigma_0\right) - \frac{1}{2}d \\
&+ \frac{1}{2}(\mu_1 - \mu_0)^\top \Sigma_1^{-1}(\mu_1 - \mu_0) \\
&+ \frac{1}{2}\ln|\det \Sigma_1| - \frac{1}{2}\ln|\det \Sigma_0|.
\end{aligned}
\tag{63}
$$

**Corollary D.2.** *Let $\mathcal{N}(\mu_0, \Sigma_0)$, $\mathcal{N}(\mu_1, \Sigma_1)$, and $\mathcal{N}(\mu_2, \Sigma_2)$ be three Gaussian distributions of dimensionality $d$ with $\Sigma_0, \Sigma_1,$ and $\Sigma_2 \succ 0$. Then,*

$$
\begin{aligned}
&KL\left(\mathcal{N}(\mu_0, \Sigma_0) \,\|\, \mathcal{N}(\mu_1, \Sigma_1)\right) \\
&+ KL\left(\mathcal{N}(\mu_0, \Sigma_0) \,\|\, \mathcal{N}(\mu_2, \Sigma_2)\right) \\
&= \operatorname{tr}\left(\left(\frac{1}{2}\Sigma_1^{-1} + \frac{1}{2}\Sigma_2^{-1}\right)\Sigma_0\right) - d \\
&+ \frac{1}{2}(\mu_1 - \mu_0)^\top \Sigma_1^{-1}(\mu_1 - \mu_0) + \frac{1}{2}(\mu_2 - \mu_0)^\top \Sigma_2^{-1}(\mu_2 - \mu_0) \\
&+ \frac{1}{2}\ln|\det \Sigma_1| + \frac{1}{2}\ln|\det \Sigma_2| - \ln|\det \Sigma_0|
\end{aligned}
$$

For any dataset $\mathcal{D}$, let $\hat{z}_\mathcal{D}$ be the maximum likelihood estimate and $-S_\mathcal{D}^{-1}$ be the Hessian evaluated at the maximum likelihood estimate $\nabla_z^2 \log p(\mathcal{D}|\hat{z}_\mathcal{D})$, such that,

$$
\hat{z}_\mathcal{D} = \operatorname*{argmax}_z \log p(\mathcal{D}|z), \qquad \text{and} \qquad S_\mathcal{D}^{-1} = -\nabla_z^2 \log p(\mathcal{D}|\hat{z}_\mathcal{D}).
\tag{64}
$$

The idea behind eq. (3) is to use the Berstein-von Misses theorem, also known as the Bayesian central limit theorem. While there are several variants of this theorem, Theorem 10.1 from Vaart (1998) suffices for our usecase. Note that Theorem 10.1 states the result in terms of the true parameters. For the extension to the maximum-likelihood estimate, see the discussion after Lemma 10.3 (Vaart, 1998).

Under eq. (3), we essentially assume that the posterior $p(z|\mathcal{D}) \approx \mathcal{N}(z_{\text{MLE}}, \frac{1}{|\mathcal{D}|}I^{-1}(z_{\text{MLE}}))$, where $I$ is the estimate of the Fisher information matrix evaluated at the maximum likelihood estimate. Since the estimate $I^{-1}(z_{\text{MLE}}) = -(\frac{1}{|\mathcal{D}|}\nabla_z^2 \log p(\mathcal{D}|z_{\text{MLE}}))^{-1}$, the $|\mathcal{D}|$ cancels out and we get

$$
p(z|\mathcal{D}) \approx \mathcal{N}\left(z|\hat{z}_\mathcal{D}, S_\mathcal{D}\right),
\tag{65}
$$
$$
p(z|\mathcal{D} + \mathcal{D}^*) \approx \mathcal{N}\left(z|\hat{z}_{\mathcal{D}+\mathcal{D}^*}, S_{\mathcal{D}+\mathcal{D}^*}\right), \text{ and}
\tag{66}
$$
$$
p(z|\mathcal{D} + 2\mathcal{D}^*) \approx \mathcal{N}\left(z|\hat{z}_{\mathcal{D}+2\mathcal{D}^*}, S_{\mathcal{D}+2\mathcal{D}^*}\right).
\tag{67}
$$

Now, assuming that the posteriors can be approximated by their corresponding Gaussian approximations under Bayesian CLT is not sufficient to guarantee that the KL divergence between the original posteriors will also converge to the KL divergence between the Gaussian approximations. Convergence of KL divergences requires additional assumptions (to be precise, we need to additionally assume absolute continuity and uniform integrability, see section D.1 for discussion). For

simplicity, we side-step the convergence of KL-divergence discussion and directly analyze the expression for the sum of KL-divergences between the Bayesian CLT approximations. Using Corollary D.2, we can simplify as follows.

$$
\begin{aligned}
&\mathrm{KL}\left(\mathcal{N}\left(\hat{z}_{\mathcal{D}+\mathcal{D}^*}, S_{\mathcal{D}+\mathcal{D}^*}\right) \,\|\, \mathcal{N}\left(\hat{z}_{\mathcal{D}}, S_{\mathcal{D}}\right)\right) \\
&\quad + \mathrm{KL}\left(\mathcal{N}\left(\hat{z}_{\mathcal{D}+\mathcal{D}^*}, S_{\mathcal{D}+\mathcal{D}^*}\right) \,\|\, \mathcal{N}\left(\hat{z}_{\mathcal{D}+2\mathcal{D}^*}, S_{\mathcal{D}+2\mathcal{D}^*}\right)\right) \\
&= \mathrm{tr}\left(\left(\left(\frac{1}{2}S_{\mathcal{D}}^{-1} + \frac{1}{2}S_{\mathcal{D}+2\mathcal{D}^*}^{-1}\right) S_{\mathcal{D}+\mathcal{D}^*}\right) - d\right. \\
&\quad + \frac{1}{2}(\hat{z}_{\mathcal{D}} - \hat{z}_{\mathcal{D}+\mathcal{D}^*})^\top S_{\mathcal{D}}^{-1}(\hat{z}_{\mathcal{D}} - \hat{z}_{\mathcal{D}+\mathcal{D}^*}) + \frac{1}{2}(\hat{z}_{\mathcal{D}+2\mathcal{D}^*} - \hat{z}_{\mathcal{D}+\mathcal{D}^*})^\top S_{\mathcal{D}+2\mathcal{D}^*}^{-1}(\hat{z}_{\mathcal{D}+2\mathcal{D}^*} - \hat{z}_{\mathcal{D}+\mathcal{D}^*}) \\
&\quad + \frac{1}{2}\ln|\det S_{\mathcal{D}}| + \frac{1}{2}\ln|\det S_{\mathcal{D}+2\mathcal{D}^*}| - \ln|\det S_{\mathcal{D}+\mathcal{D}^*}|.
\end{aligned}
\tag{68}
$$

The above expression is terse and hides the relationship of SNR to the dimensionality of the problem and the relative size of the datasets. To bring out these relationships, we make the simplifying assumption that the datasets $\mathcal{D}$, $\mathcal{D}+\mathcal{D}^*$, and $\mathcal{D}+2\mathcal{D}^*$ are similar enough such that the maximum likelihood estimate and the estimate of the Fisher Information matrix is similar under the three datasets. Mathematically, we assume that

$$
\hat{z}_{\mathcal{D}} \approx \hat{z}_{\mathcal{D}+\mathcal{D}^*} \approx \hat{z}_{\mathcal{D}+2\mathcal{D}^*}
\tag{69}
$$

and empirical Fisher Information matrix (or the scaled Hessian) is given by

$$
\frac{1}{|\mathcal{D}|}S_{\mathcal{D}}^{-1} \approx \frac{1}{|\mathcal{D}+\mathcal{D}^*|}S_{\mathcal{D}+\mathcal{D}^*}^{-1} \approx \frac{1}{|\mathcal{D}+2\mathcal{D}^*|}S_{\mathcal{D}+2\mathcal{D}^*}^{-1}.
\tag{70}
$$

Substituting from eqs. (69) and (70) into eq. (68), and simplifying as in section D.2, we get

$$
\begin{aligned}
&\mathrm{KL}\left(\mathcal{N}\left(\hat{z}_{\mathcal{D}+\mathcal{D}^*}, S_{\mathcal{D}+\mathcal{D}^*}\right) \,\|\, \mathcal{N}\left(\hat{z}_{\mathcal{D}}, S_{\mathcal{D}}\right)\right) \\
&\quad + \mathrm{KL}\left(\mathcal{N}\left(\hat{z}_{\mathcal{D}+\mathcal{D}^*}, S_{\mathcal{D}+\mathcal{D}^*}\right) \,\|\, \mathcal{N}\left(\hat{z}_{\mathcal{D}+2\mathcal{D}^*}, S_{\mathcal{D}+2\mathcal{D}^*}\right)\right) \\
&\approx d\log\frac{|\mathcal{D}+\mathcal{D}^*|}{\sqrt{|\mathcal{D}|\,|\mathcal{D}+2\mathcal{D}^*|}}.
\end{aligned}
\tag{71}
$$

Finally, plugging the KL-divergences from eq. (71) into the definition of $\delta$ in eq. (3), we get the result

$$
\delta \approx \frac{1}{2}d\log\frac{|\mathcal{D}+\mathcal{D}^*|}{\sqrt{|\mathcal{D}|\,|\mathcal{D}+2\mathcal{D}^*|}} = \frac{1}{2}d\log\frac{1 + |\mathcal{D}^*|/|\mathcal{D}|}{\sqrt{1 + 2\,|\mathcal{D}^*|/|\mathcal{D}|}}.
\tag{72}
$$

As we can see, the assumption about the similarity of the empirical Fisher Information matrix and the MLE is necessary to bring out the dependence of SNR on dimensionality and the relative size of datasets. This assumption over the datasets is relatively strong and the main reason we keep the above result informal. Note that the middle term in the above equation shows that $\delta$ is positive—the quantity inside the logarithm is larger than one since $|\mathcal{D}+\mathcal{D}^*|$ is the arithmetic mean of $|\mathcal{D}|$ and $|\mathcal{D}+2\mathcal{D}^*|$ which is always larger than the geometric mean $\sqrt{|\mathcal{D}|\,|\mathcal{D}+2\mathcal{D}^*|}$. The right term clarifies that only the dimensionality and ratio of $|\mathcal{D}|$ and $|\mathcal{D}^*|$ matters.

### D.1. Note on convergence of KL-divergence

Although the Bayesian CLT or (Bernstein–von Mises theorem) provides convergence in total variation between each posterior distribution and its normal approximation, this convergence alone is not sufficient to guarantee the convergence of the KL-divergence between two sequences of posterior distributions. KL divergence is highly sensitive to differences in how probability mass is allocated, especially in regions where one distribution assigns significant probability and the other does not. This sensitivity means that even with total variation convergence, the KL divergence between the sequences can diverge or fail to converge to the KL divergence between the limits.

To ensure convergence of the KL divergence, additional assumptions of absolute continuity and uniform integrability are required. Absolute continuity guarantees that wherever one distribution assigns positive probability, the other does

too—preventing the KL divergence from becoming infinite due to zero probabilities in the denominator of the density ratio. Uniform integrability of the log-density ratios is also required; it controls the contributions to the KL divergence from regions where the density ratios become extreme (either very large or very small). By ensuring that these extreme values do not disproportionately influence the KL divergence, it prevents divergence caused by small probabilities or heavy tails.

Collectively, these assumptions ensure that the differences between the distributions are well-behaved for the KL divergence to converge. We skip the formal proof as it is not the focus of this paper.

### D.2. Note for the simplification from eq. (68) to eq. (71)

When the datasets $\mathcal{D}^*$ and $\mathcal{D}$ have the matching mean statistics, we have the relations in eqs. (69) and (70). Under eq. (69), the quadratic terms in eq. (68) are zero. We can simplify the term involving trace as follows:

$$
\begin{aligned}
&\mathrm{tr}\left(\left(\frac{1}{2}S_{\mathcal{D}}^{-1} + \frac{1}{2}S_{\mathcal{D}+2\mathcal{D}^*}^{-1}\right)S_{\mathcal{D}+\mathcal{D}^*}\right) \\
&= \frac{1}{2}\,\mathrm{tr}\left(S_{\mathcal{D}}^{-1}S_{\mathcal{D}+\mathcal{D}^*}\right) + \frac{1}{2}\,\mathrm{tr}\left(S_{\mathcal{D}+2\mathcal{D}^*}^{-1}S_{\mathcal{D}+\mathcal{D}^*}\right) \\
&\overset{(a)}{\approx} \frac{1}{2}\,\mathrm{tr}\left(S_{\mathcal{D}}^{-1}\left(\frac{|\mathcal{D}+\mathcal{D}^*|}{|\mathcal{D}|}S_{\mathcal{D}}^{-1}\right)^{-1}\right) + \frac{1}{2}\,\mathrm{tr}\left(\left(\frac{|\mathcal{D}+2\mathcal{D}^*|}{|\mathcal{D}|}S_{\mathcal{D}}^{-1}\right)\left(\frac{|\mathcal{D}+\mathcal{D}^*|}{|\mathcal{D}|}S_{\mathcal{D}}^{-1}\right)^{-1}\right) \\
&= \frac{1}{2}\frac{|\mathcal{D}|}{|\mathcal{D}+\mathcal{D}^*|}\,\mathrm{tr}\left(S_{\mathcal{D}}^{-1}S_{\mathcal{D}}\right) + \frac{1}{2}\frac{|\mathcal{D}+2\mathcal{D}^*|}{|\mathcal{D}|}\frac{|\mathcal{D}|}{|\mathcal{D}+\mathcal{D}^*|}\,\mathrm{tr}\left(S_{\mathcal{D}}^{-1}S_{\mathcal{D}}\right) \\
&= \frac{1}{2}\frac{|\mathcal{D}|}{|\mathcal{D}+\mathcal{D}^*|}d + \frac{1}{2}\frac{|\mathcal{D}+2\mathcal{D}^*|}{|\mathcal{D}+\mathcal{D}^*|}d \\
&\overset{(b)}{=} \frac{\frac{1}{2}|\mathcal{D}| + \frac{1}{2}|\mathcal{D}+2\mathcal{D}^*|}{|\mathcal{D}+\mathcal{D}^*|}d \\
&= \frac{|\mathcal{D}+\mathcal{D}^*|}{|\mathcal{D}+\mathcal{D}^*|}d \\
&= d,
\end{aligned}
$$

where (a) follows from the relation in eq. (70) and (b) follows from the multiset notation (Costa, 2021).

Therefore, the first and the second term ($d$ and $-d$) in eq. (68) cancel out, and the only remaining terms are the ones involving the logarithms of the determinants of the covariance matrices. These remaining terms can be simplified as follows:

$$
\begin{aligned}
&\frac{1}{2}\ln\det\left(S_{\mathcal{D}}\right) + \frac{1}{2}\ln\det\left(S_{\mathcal{D}+2\mathcal{D}^*}\right) - \ln\det\left(S_{\mathcal{D}+\mathcal{D}^*}\right) \\
&\overset{(c)}{\approx} \frac{1}{2}\ln\det\left(S_{\mathcal{D}}\right) + \frac{1}{2}\ln\det\left(\frac{|\mathcal{D}|}{|\mathcal{D}+2\mathcal{D}^*|}S_{\mathcal{D}}\right) - \ln\det\left(\frac{|\mathcal{D}|}{|\mathcal{D}+\mathcal{D}^*|}S_{\mathcal{D}}\right) \\
&\overset{(d)}{=} \frac{1}{2}\ln\det\left(S_{\mathcal{D}}\right) + \frac{1}{2}\ln\det\left(S_{\mathcal{D}}\right) + \frac{d}{2}\log\left(\frac{|\mathcal{D}|}{|\mathcal{D}+2\mathcal{D}^*|}\right) - \ln\det\left(S_{\mathcal{D}}\right) - d\log\left(\frac{|\mathcal{D}|}{|\mathcal{D}+\mathcal{D}^*|}\right) \\
&= \frac{d}{2}\log\left(\frac{|\mathcal{D}|}{|\mathcal{D}+2\mathcal{D}^*|}\right) - d\log\left(\frac{|\mathcal{D}|}{|\mathcal{D}+\mathcal{D}^*|}\right) \\
&\overset{(f)}{=} d\left(\log|\mathcal{D}+\mathcal{D}^*| - \frac{1}{2}\log|\mathcal{D}| - \frac{1}{2}\log|\mathcal{D}+2\mathcal{D}^*|\right) \\
&= d\log\frac{|\mathcal{D}+\mathcal{D}^*|}{\sqrt{|\mathcal{D}|\,|\mathcal{D}+2\mathcal{D}^*|}},
\end{aligned}
$$

where (f) follows from eq. (70); (d) follows from $\log\det(aX) = d\log a + \log\det X$ for any non-negative scalar $a$; this gives the final result in eq. (71); and (c) follows from simple algebraic manipulations.

# E. Proof for Theorem 2.2

This section presents a proof for theorem 2.2 about the SNR in the linear regression model. First, we formally define the Bayesian linear regression models in definition E.1. We then restate the assumptions in theorem 2.2 for convenience. Lemma E.2 presents the expression for the posterior distribution where we consider both training and test data under the assumption B1. Lemma E.3 presents the KL divergence between two posterior distributions with test data under the assumptions B1 and B2. Finally, we use the KL expression from lemmas E.2 and E.3 to prove theorem 2.2.

**Definition E.1** (Bayesian Linear Regression Model). Consider the linear regression model with a Gaussian likelihood such that

$$p(y_\mathcal{D}|z) = \mathcal{N}(y_\mathcal{D}|X_\mathcal{D}z, \sigma^2 I). \tag{73}$$

where $y_\mathcal{D} \in \mathbb{R}^{|\mathcal{D}|}$ is the response vector, $X_\mathcal{D} \in \mathbb{R}^{|\mathcal{D}| \times d}$ is feature matrix, and $\sigma^2$ is the variance.

The conjugate prior is a Gaussian distribution such that

$$p(z) = \mathcal{N}(z|\mu_0, \Sigma_0) \tag{74}$$

where $\mu_0$ is the mean and $\Sigma_0$ is the covariance. Then, the posterior distribution is given by

$$p(z|y_\mathcal{D}) = \mathcal{N}(z|\mu_\mathcal{D}, \Sigma_\mathcal{D}), \tag{75}$$

where

$$\Sigma_\mathcal{D} = \left( \frac{1}{\sigma^2} X_\mathcal{D}^\top X_\mathcal{D} + \Sigma_0^{-1} \right)^{-1} \quad \text{and} \quad \mu_\mathcal{D} = \Sigma_\mathcal{D} \left( \frac{1}{\sigma^2} X_\mathcal{D}^\top y_\mathcal{D} + \Sigma_0^{-1} \mu_0 \right). \tag{76}$$

**Lemma E.2.** *Let $p$ be the Bayesian linear regression model from definition E.1. Let B1 and B2 hold. Let $c$ be a non-negative integer. Then,*

$$p(z|\mathcal{D} + c\mathcal{D}^*) = \mathcal{N}(z|\mu_{\mathcal{D}+c\mathcal{D}^*}, \Sigma_{\mathcal{D}+c\mathcal{D}^*}), \tag{77}$$

*where as $|\mathcal{D}| \to \infty$,*

$$\Sigma_{\mathcal{D}+c\mathcal{D}^*} \to \frac{1}{c+1} \left( \frac{1}{\sigma^2} X_\mathcal{D}^\top X_\mathcal{D} \right)^{-1} \quad \text{and} \quad \mu_{\mathcal{D}+c\mathcal{D}^*} \to \left( X_\mathcal{D}^\top X_\mathcal{D} \right)^{-1} X_\mathcal{D}^\top \left( y_\mathcal{D} + \frac{c}{c+1} \Delta \right). \tag{78}$$

*Proof.* We first massage the expressions for the covariance and the mean of the posterior distribution such that we can use the B1 and B2.

$$\Sigma_\mathcal{D}$$

$$= \left( \frac{1}{\sigma^2} X_\mathcal{D}^\top X_\mathcal{D} + \Sigma_0^{-1} \right)^{-1} \tag{79}$$

$$= \left( (X_\mathcal{D}^\top X_\mathcal{D}) \left( \frac{1}{\sigma^2} I + (X_\mathcal{D}^\top X_\mathcal{D})^{-1} \Sigma_0^{-1} \right) \right)^{-1} \tag{80}$$

Based on the B1, we have

$$X_{\mathcal{D}+c\mathcal{D}^*}^\top X_{\mathcal{D}+c\mathcal{D}^*} = \begin{bmatrix} X_\mathcal{D} \\ X_\mathcal{D} \\ \vdots \\ X_\mathcal{D} \end{bmatrix}^\top \begin{bmatrix} X_\mathcal{D} \\ X_\mathcal{D} \\ \vdots \\ X_\mathcal{D} \end{bmatrix} = (c+1) X_\mathcal{D}^\top X_\mathcal{D} \tag{81}$$

Plugging eq. (81) into eq. (80) gives

$$\Sigma_{\mathcal{D}+c\mathcal{D}^*} = \left( ((c+1) X_\mathcal{D}^\top X_\mathcal{D}) \left( \frac{1}{\sigma^2} I + ((c+1) X_\mathcal{D}^\top X_\mathcal{D})^{-1} \Sigma_0^{-1} \right) \right)^{-1}. \tag{82}$$

A similar massaging of expressions for the mean gives us

$$\mu_{\mathcal{D}}$$

$$= \Sigma_{\mathcal{D}} \left( \frac{1}{\sigma^2} X_{\mathcal{D}}^\top y_{\mathcal{D}} + \Sigma_0^{-1} \mu_0 \right) \tag{83}$$

$$= \left( \frac{1}{\sigma^2} X_{\mathcal{D}}^\top X_{\mathcal{D}} + \Sigma_0^{-1} \right)^{-1} \left( \frac{1}{\sigma^2} X_{\mathcal{D}}^\top y_{\mathcal{D}} + \Sigma_0^{-1} \mu_0 \right) \tag{84}$$

$$= \left( (X_{\mathcal{D}}^\top X_{\mathcal{D}}) \left( \frac{1}{\sigma^2} I + (X_{\mathcal{D}}^\top X_{\mathcal{D}})^{-1} \Sigma_0^{-1} \right) \right)^{-1} \left( \frac{1}{\sigma^2} X_{\mathcal{D}}^\top y_{\mathcal{D}} + \Sigma_0^{-1} \mu_0 \right) \tag{85}$$

$$= \left( \frac{1}{\sigma^2} I + (X_{\mathcal{D}}^\top X_{\mathcal{D}})^{-1} \Sigma_0^{-1} \right)^{-1} (X_{\mathcal{D}}^\top X_{\mathcal{D}})^{-1} \left( \frac{1}{\sigma^2} X_{\mathcal{D}}^\top y_{\mathcal{D}} + \Sigma_0^{-1} \mu_0 \right) \tag{86}$$

$$= \left( \frac{1}{\sigma^2} I + (X_{\mathcal{D}}^\top X_{\mathcal{D}})^{-1} \Sigma_0^{-1} \right)^{-1} \left( \frac{1}{\sigma^2} (X_{\mathcal{D}}^\top X_{\mathcal{D}})^{-1} X_{\mathcal{D}}^\top y_{\mathcal{D}} + (X_{\mathcal{D}}^\top X_{\mathcal{D}})^{-1} \Sigma_0^{-1} \mu_0 \right) \tag{87}$$

Now, based on the B1, we have eq. (88).

$$X_{\mathcal{D}+c\mathcal{D}^*}^\top y_{\mathcal{D}+c\mathcal{D}^*} = \begin{bmatrix} X_{\mathcal{D}} \\ X_{\mathcal{D}} \\ \vdots \\ X_{\mathcal{D}} \end{bmatrix}^\top \begin{bmatrix} y_{\mathcal{D}} \\ y_{\mathcal{D}} + \Delta \\ \vdots \\ y_{\mathcal{D}} + \Delta \end{bmatrix} = X_{\mathcal{D}}^\top ((c+1)y_{\mathcal{D}} + c\Delta) \tag{88}$$

Plugging eq. (81) and eq. (88) into eq. (87) gives

$$\mu_{\mathcal{D}+c\mathcal{D}^*} = \left( \frac{1}{\sigma^2} I + ((c+1)X_{\mathcal{D}}^\top X_{\mathcal{D}})^{-1} \Sigma_0^{-1} \right)^{-1}$$

$$\left( \frac{1}{\sigma^2} ((c+1)X_{\mathcal{D}}^\top X_{\mathcal{D}})^{-1} X_{\mathcal{D}}^\top ((c+1)y_{\mathcal{D}} + c\Delta) + ((c+1)X_{\mathcal{D}}^\top X_{\mathcal{D}})^{-1} \Sigma_0^{-1} \mu_0 \right), \tag{89}$$

Now, taking limits (as $|\mathcal{D}| \to \infty$ implies $(X_{\mathcal{D}}^\top X_{\mathcal{D}})^{-1} \Sigma_0^{-1} \to 0$ from B2) on both sides of eq. (89) leads to simplification of terms inside the brackets and gives us

$$\lim_{|\mathcal{D}| \to \infty} \Sigma_{\mathcal{D}+c\mathcal{D}^*}$$

$$= \lim_{|\mathcal{D}| \to \infty} \left( ((c+1)X_{\mathcal{D}}^\top X_{\mathcal{D}}) \left( \frac{1}{\sigma^2} I + ((c+1)X_{\mathcal{D}}^\top X_{\mathcal{D}})^{-1} \Sigma_0^{-1} \right) \right)^{-1} \tag{90}$$

$$= \frac{1}{c+1} \left( \frac{1}{\sigma^2} X_{\mathcal{D}}^\top X_{\mathcal{D}} \right)^{-1}. \tag{91}$$

Similarly, taking the limits on both sides of eq. (87) leads to simplification of terms inside the brackets (under the obvious assumption that $\mu_0$ is finite) and gives us

$$\lim_{|\mathcal{D}| \to \infty} \mu_{\mathcal{D}+c\mathcal{D}^*}$$

$$= \lim_{|\mathcal{D}| \to \infty} \left( \frac{1}{\sigma^2} I + ((c+1)X_{\mathcal{D}}^\top X_{\mathcal{D}})^{-1} \Sigma_0^{-1} \right)^{-1}$$

$$\left( \frac{1}{\sigma^2} ((c+1)X_{\mathcal{D}}^\top X_{\mathcal{D}})^{-1} X_{\mathcal{D}}^\top ((c+1)y_{\mathcal{D}} + c\Delta) + ((c+1)X_{\mathcal{D}}^\top X_{\mathcal{D}})^{-1} \Sigma_0^{-1} \mu_0 \right), \tag{92}$$

$$= (X_{\mathcal{D}}^\top X_{\mathcal{D}})^{-1} X_{\mathcal{D}}^\top (y_{\mathcal{D}} + \frac{c}{c+1} \Delta). \tag{93}$$

$$\square$$

**Lemma E.3.** *Let $p$ be the Bayesian linear regression model from definition E.1. Let B1 and B2 hold. Let $\alpha$ and $\beta$ be two non-negative integers. Then, as $|\mathcal{D}| \to \infty$,*

$$KL\left(p(z|\mathcal{D} + \alpha\mathcal{D}^*) \,\|\, p(z|\mathcal{D} + \beta\mathcal{D}^*)\right) \to \frac{1}{2}\left(k_{\alpha,\beta}d + \Delta^\top M_{\alpha,\beta}\Delta\right), \tag{94}$$

*where $k_{\alpha,\beta}$ is a positive constant and $M_{\alpha,\beta}$ is a positive definite matrix such that*

$$k_{\alpha,\beta} = \frac{\beta+1}{\alpha+1} + \log\frac{\alpha+1}{\beta+1} - 1 \quad \text{and} \quad M_{\alpha,\beta} = \frac{(\beta-\alpha)^2}{(\alpha+1)^2(\beta+1)}\frac{1}{\sigma^2}X_\mathcal{D}\left(X_\mathcal{D}^\top X_\mathcal{D}\right)^{-1}X_\mathcal{D}^\top. \tag{95}$$

*Proof.* The result follows directly from plugging the mean and the covariance into the expression for KL divergence between the two Gaussians, taking limits, and following simple algebraic manipulations. We know that

$$KL\left(\mathcal{N}(\mu_1, \Sigma_1) \,\|\, \mathcal{N}(\mu_2, \Sigma_2)\right) = \frac{1}{2}\left(\text{tr}(\Sigma_2^{-1}\Sigma_1) + \log\frac{\det\Sigma_2}{\det\Sigma_1} - d + (\mu_2-\mu_1)^\top\Sigma_2^{-1}(\mu_2-\mu_1)\right). \tag{96}$$

Collecting the transpose terms, plugging in the covariance expressions from eq. (82) for the distributions $p(z|\mathcal{D} + \alpha\mathcal{D}^*)$ and $p(z|\mathcal{D} + \beta\mathcal{D}^*)$, and taking the limits (as $|\mathcal{D}| \to \infty$ implies $\left(X_\mathcal{D}^\top X_\mathcal{D}\right)^{-1}\Sigma_0^{-1} \to 0$ from B2) we get

$$\lim_{|\mathcal{D}|\to\infty}\text{tr}(\Sigma_2^{-1}\Sigma_1) = \frac{\beta+1}{\alpha+1}d. \tag{97}$$

See section E.2.1 for details of the calculations. Collecting the determinant terms, plugging the covariance expressions from eq. (82) for the distributions $p(z|\mathcal{D} + \alpha\mathcal{D}^*)$ and $p(z|\mathcal{D} + \beta\mathcal{D}^*)$, and taking the limits we get

$$\lim_{|\mathcal{D}|\to\infty}\log\frac{\det\Sigma_2}{\det\Sigma_1} = \left(\log\frac{\alpha+1}{\beta+1}\right)d. \tag{98}$$

See section E.2.2 for details of the calculations.

Finally, plugging in the expressions for the covariance and the mean from eqs. (82) and (89) for $p(z|\mathcal{D} + \alpha\mathcal{D}^*)$ and $p(z|\mathcal{D} + \beta\mathcal{D}^*)$ in the quadratic term, and taking the limits we get

$$\lim_{|\mathcal{D}|\to\infty}(\mu_2-\mu_1)^\top\Sigma_2^{-1}(\mu_2-\mu_1) = \frac{(\beta-\alpha)^2}{(\alpha+1)^2(\beta+1)}\frac{1}{\sigma^2}\Delta^\top X_\mathcal{D}\left(X_\mathcal{D}^\top X_\mathcal{D}\right)^{-1}X_\mathcal{D}^\top\Delta. \tag{99}$$

See section E.2.3 for details of the calculations. Plugging the aforementioned limit results back into the KL-divergence expression and identifying the correct coefficients gives the result. $\qquad\square$

**Theorem E.4** (Repeated for convenience)**.** *Let $p(z) = \mathcal{N}(z|\mu_0, \Sigma_0)$ and $p(y_\mathcal{D}|z) = \mathcal{N}(y_\mathcal{D}|X_\mathcal{D}z, \sigma^2 I)$, where $y_\mathcal{D} \in \mathbb{R}^{|\mathcal{D}|}$ are the responses, $X_\mathcal{D} \in \mathbb{R}^{|\mathcal{D}|\times d}$ are the features, $z \in \mathbb{R}^d$ are the weights, and $\sigma^2$ is the known variance. Let $\delta$ be as in eq. (3).*

- *B1: The test features $X_{\mathcal{D}^*}$ consist of $m$ copies of $X_\mathcal{D}$ and responses $y_{\mathcal{D}^*}$ consist of $m$ copies of $y_\mathcal{D} + \Delta$ where $\Delta$ is the mismatch vector of size $|\mathcal{D}|$.*

- *B2: For a sequence of increasingly large training datasets, posterior is dominated by data. Mathematically, as $|\mathcal{D}| \to \infty$, $\left(X_\mathcal{D}^\top X_\mathcal{D}\right)^{-1}\Sigma_0^{-1} \to 0$.*

*Assume B1 and B2. Then, $\lim_{|\mathcal{D}|\to\infty}|\delta - \delta_{\text{lim}}| = 0$ and*

$$\delta_{\text{lim}} = \frac{d}{2}\log\frac{1+m}{\sqrt{1+2m}}$$
$$+ \frac{1}{2\sigma^2}\frac{m^2}{2m^2+3m+1}\Delta^\top X_\mathcal{D}\left(X_\mathcal{D}^\top X_\mathcal{D}\right)^{-1}X_\mathcal{D}^\top\Delta. \tag{100}$$

*Proof.* $\delta$ can be written as an average of the KL-divergences between the posteriors $p(z|\mathcal{D} + \mathcal{D}^*)$ and $p(z|\mathcal{D})$ and between the posteriors $p(z|\mathcal{D} + \mathcal{D}^*)$ and $p(z|\mathcal{D} + 2\mathcal{D}^*)$. From the expressions of KL divergences in lemma E.3, we get

$$\lim_{|\mathcal{D}| \to \infty} \delta = \frac{1}{2} \left[ \frac{1}{2} \left( k_{m,0}d + \Delta^\top M_{m,0}\Delta \right) + \frac{1}{2} \left( k_{m,2m}d + \Delta^\top M_{m,2m}\Delta \right) \right], \tag{101}$$

$$= \frac{1}{2} \left[ \frac{1}{2} \left( k_{m,0} + k_{m,2m} \right) d + \frac{1}{2} \Delta^\top \left( M_{m,0} + M_{m,2m} \right) \Delta \right], \tag{102}$$

Simplifying the expressions, we get

$$k_{m,0} + k_{m,2m} \tag{103}$$

$$= \log \frac{1+m}{1} + \frac{1}{1+m} - 1 + \log \frac{1+m}{1+2m} + \frac{1+2m}{1+m} - 1 \tag{104}$$

$$= \log \frac{1+m}{1} + \log \frac{1+m}{1+2m} \tag{105}$$

$$= \log \frac{(1+m)^2}{1+2m} \tag{106}$$

$$= 2 \log \frac{1+m}{\sqrt{1+2m}} \tag{107}$$

$$\tag{108}$$

and

$$M_{m,0} + M_{m,2m} \tag{109}$$

$$= \frac{m^2}{(m+1)^2} \frac{1}{\sigma^2} X_\mathcal{D} \left( X_\mathcal{D}^\top X_\mathcal{D} \right)^{-1} X_\mathcal{D}^\top + \frac{m^2}{(m+1)^2 (2m+1)} \frac{1}{\sigma^2} X_\mathcal{D} \left( X_\mathcal{D}^\top X_\mathcal{D} \right)^{-1} X_\mathcal{D}^\top \tag{110}$$

$$= \frac{m^2}{(m+1)^2} \frac{1}{\sigma^2} X_\mathcal{D} \left( X_\mathcal{D}^\top X_\mathcal{D} \right)^{-1} X_\mathcal{D}^\top \left( 1 + \frac{1}{2m+1} \right) \tag{111}$$

$$= \frac{m^2}{(m+1)^2} \frac{2(m+1)}{2m+1} \frac{1}{\sigma^2} X_\mathcal{D} \left( X_\mathcal{D}^\top X_\mathcal{D} \right)^{-1} X_\mathcal{D}^\top \tag{112}$$

$$= \frac{2m^2}{(m+1)(2m+1)} \frac{1}{\sigma^2} X_\mathcal{D} \left( X_\mathcal{D}^\top X_\mathcal{D} \right)^{-1} X_\mathcal{D}^\top \tag{113}$$

$$\tag{114}$$

Plugging these back in eq. (102) we get the results. $\qquad\square$

### E.1. Bounding the $\delta_{\mathrm{lim}}$ in theorem 2.2

$\delta_{\mathrm{lim}}$ can be bounded by bounding three individual terms. First, $\log(1+m)/\sqrt{1+2m}$ is lower-bounded by $d/4 \log(m/2)$ and upper-bounded by $d/4 \log(m/2 + 1)$. Second, $m^2 / \left( 2m^2 + 3m + 1 \right)$ is lower-bounded by $1/6$ and upper-bounded by $1/2$. Third, we have

$$\Delta^\top X_\mathcal{D} \left( X_\mathcal{D}^\top X_\mathcal{D} \right)^{-1} X_\mathcal{D}^\top \Delta = \Delta^\top U U^\top \Delta \tag{115}$$

where $U$ is the left singular matrix of $X_\mathcal{D}$ containing $d$ singular left vectors. Then, from the properties of the left-singular vectors, $\|U^\top \Delta\|_2^2$ terms is lower-bounded by $0$ and upper-bounded by $\|\Delta\|_2^2$. Combining these bounds, we get the bounds in eq. (7).

### E.2. Note for the calculation of the KL terms in lemma E.3

We will consider the calculations for the three terms in KL divergence between two Gaussian distributions.

### E.2.1. TRANSPOSE TERMS

The first term we focus on is the one involving transpose.

$$\text{tr}(\Sigma_2^{-1}\Sigma_1) \tag{116}$$

$$= \text{tr}(\Sigma_{\mathcal{D}+\beta\mathcal{D}^*}^{-1}\Sigma_{\mathcal{D}+\alpha\mathcal{D}^*}) \tag{117}$$

$$= \text{tr}\left[\left((\beta+1)X_\mathcal{D}^\top X_\mathcal{D}\right)\left(\frac{1}{\sigma^2}I + \left((\beta+1)X_\mathcal{D}^\top X_\mathcal{D}\right)^{-1}\Sigma_0^{-1}\right)\right.$$
$$\left.\left(\left((\alpha+1)X_\mathcal{D}^\top X_\mathcal{D}\right)\left(\frac{1}{\sigma^2}I + \left((\alpha+1)X_\mathcal{D}^\top X_\mathcal{D}\right)^{-1}\Sigma_0^{-1}\right)\right)^{-1}\right] \tag{118}$$

$$= \text{tr}\left[\left((\beta+1)X_\mathcal{D}^\top X_\mathcal{D}\right)\left(\frac{1}{\sigma^2}I + \left((\beta+1)X_\mathcal{D}^\top X_\mathcal{D}\right)^{-1}\Sigma_0^{-1}\right)\right.$$
$$\left.\frac{1}{\alpha+1}\left(X_\mathcal{D}^\top X_\mathcal{D}\right)^{-1}\left(\frac{1}{\sigma^2}I + \left((\alpha+1)X_\mathcal{D}^\top X_\mathcal{D}\right)^{-1}\Sigma_0^{-1}\right)^{-1}\right] \tag{119}$$

$$= \text{tr}\left[\frac{\beta+1}{\alpha+1}\left(\frac{1}{\sigma^2}I + \left((\beta+1)X_\mathcal{D}^\top X_\mathcal{D}\right)^{-1}\Sigma_0^{-1}\right)\left(\frac{1}{\sigma^2}I + \left((\alpha+1)X_\mathcal{D}^\top X_\mathcal{D}\right)^{-1}\Sigma_0^{-1}\right)^{-1}\right] \tag{120}$$

$$= \frac{\beta+1}{\alpha+1}\,\text{tr}\left[\left(\frac{1}{\sigma^2}I + \left((\beta+1)X_\mathcal{D}^\top X_\mathcal{D}\right)^{-1}\Sigma_0^{-1}\right)\left(\frac{1}{\sigma^2}I + \left((\alpha+1)X_\mathcal{D}^\top X_\mathcal{D}\right)^{-1}\Sigma_0^{-1}\right)^{-1}\right] \tag{121}$$

$$\tag{122}$$

On taking limits as $|\mathcal{D}| \to \infty$ (which is equivalent to $\left(X_\mathcal{D}^\top X_\mathcal{D}\right)^{-1}\Sigma_0^{-1} \to 0$ from B2) for the above expression and doing some simple manipulations gives

$$\lim_{|\mathcal{D}|\to\infty}\text{tr}(\Sigma_2^{-1}\Sigma_1)$$
$$= \frac{\beta+1}{\alpha+1}\,\text{tr}\left[\left(\frac{1}{\sigma^2}I\right)\left(\frac{1}{\sigma^2}I\right)^{-1}\right] = \frac{\beta+1}{\alpha+1}d. \tag{123}$$

### E.2.2. DETERMINANT TERMS

The second term we consider is the one involving determinants.

$$\log\frac{\det\Sigma_2}{\det\Sigma_1} \tag{124}$$

$$= \log\det\Sigma_2 - \log\det\Sigma_1 \tag{125}$$

$$= -\log\det\left((\beta+1)X_\mathcal{D}^\top X_\mathcal{D}\right)\left(\frac{1}{\sigma^2}I + \left((\beta+1)X_\mathcal{D}^\top X_\mathcal{D}\right)^{-1}\Sigma_0^{-1}\right) \tag{126}$$

$$\quad + \log\det\left((\alpha+1)X_\mathcal{D}^\top X_\mathcal{D}\right)\left(\frac{1}{\sigma^2}I + \left((\alpha+1)X_\mathcal{D}^\top X_\mathcal{D}\right)^{-1}\Sigma_0^{-1}\right) \tag{127}$$

$$= -d\log(\beta+1) - \log\det X_\mathcal{D}^\top X_\mathcal{D} - \log\det\left(\frac{1}{\sigma^2}I + \left((\beta+1)X_\mathcal{D}^\top X_\mathcal{D}\right)^{-1}\Sigma_0^{-1}\right) \tag{128}$$

$$\quad + d\log(\alpha+1) + \log\det X_\mathcal{D}^\top X_\mathcal{D} + \log\det\left(\frac{1}{\sigma^2}I + \left((\alpha+1)X_\mathcal{D}^\top X_\mathcal{D}\right)^{-1}\Sigma_0^{-1}\right) \tag{129}$$

$$= d\log\frac{\alpha+1}{\beta+1} - \log\det\left(\frac{1}{\sigma^2}I + \left((\beta+1)X_\mathcal{D}^\top X_\mathcal{D}\right)^{-1}\Sigma_0^{-1}\right) \tag{130}$$

$$\quad + \log\det\left(\frac{1}{\sigma^2}I + \left((\alpha+1)X_\mathcal{D}^\top X_\mathcal{D}\right)^{-1}\Sigma_0^{-1}\right) \tag{131}$$

On taking limits for the above expression and doing some simple manipulations gives

$$\lim_{|\mathcal{D}|\to\infty} \log\frac{\det\Sigma_2}{\det\Sigma_1} = d\log\frac{\alpha+1}{\beta+1} \tag{132}$$

### E.2.3. QUADRATIC TERMS

Noting that the terms involved are well-behaved functions of mean and covariance, we directly plug the limit expressions for mean and covariance from Lemma E.2.

$$\lim_{|\mathcal{D}|\to\infty}(\mu_2-\mu_1)^\top\Sigma_2^{-1}(\mu_2-\mu_1) \tag{133}$$

$$= \left(\left(X_\mathcal{D}^\top X_\mathcal{D}\right)^{-1}X_\mathcal{D}^\top(y_\mathcal{D}+\frac{\beta}{\beta+1}\Delta) - \left(X_\mathcal{D}^\top X_\mathcal{D}\right)^{-1}X_\mathcal{D}^\top(y_\mathcal{D}+\frac{\alpha}{\alpha+1}\Delta)\right)^\top$$
$$(\beta+1)\left(\frac{1}{\sigma^2}X_\mathcal{D}^\top X_\mathcal{D}\right)$$
$$\left(\left(X_\mathcal{D}^\top X_\mathcal{D}\right)^{-1}X_\mathcal{D}^\top(y_\mathcal{D}+\frac{\beta}{\beta+1}\Delta) - \left(X_\mathcal{D}^\top X_\mathcal{D}\right)^{-1}X_\mathcal{D}^\top(y_\mathcal{D}+\frac{\alpha}{\alpha+1}\Delta)\right) \tag{134}$$

$$= \left(\left(X_\mathcal{D}^\top X_\mathcal{D}\right)^{-1}X_\mathcal{D}^\top\left(\frac{\beta}{\beta+1}-\frac{\alpha}{\alpha+1}\right)\Delta\right)^\top$$
$$(\beta+1)\left(\frac{1}{\sigma^2}X_\mathcal{D}^\top X_\mathcal{D}\right)$$
$$\left(\left(X_\mathcal{D}^\top X_\mathcal{D}\right)^{-1}X_\mathcal{D}^\top\left(\frac{\beta}{\beta+1}-\frac{\alpha}{\alpha+1}\right)\Delta\right) \tag{135}$$

$$= \frac{(\beta+1)(\beta-\alpha)^2}{(\alpha+1)^2(\beta+1)^2}\frac{1}{\sigma^2}\Delta^\top X_\mathcal{D}\left(X_\mathcal{D}^\top X_\mathcal{D}\right)^{-\top}\left(X_\mathcal{D}^\top X_\mathcal{D}\right)\left(X_\mathcal{D}^\top X_\mathcal{D}\right)^{-1}X_\mathcal{D}^\top\Delta \tag{136}$$

$$= \frac{(\beta-\alpha)^2}{(\alpha+1)^2(\beta+1)}\frac{1}{\sigma^2}\Delta^\top X_\mathcal{D}\left(X_\mathcal{D}^\top X_\mathcal{D}\right)^{-1}X_\mathcal{D}^\top\Delta \tag{137}$$

# F. Proof for Corollary 2.3

This section contains the proof for corollary 2.3. The overall idea of the proof is similar to the proof of theorem 2.1 in section C. Lemma F.1 provides the expression for the moments of the naive MC estimator for the conjugate exponential family models. Lemma F.2 provides the expression for the sum of KL divergences between three posterior distributions in the conjugate exponential family models, where the first distribution is the same for the two KL terms. Finally, we prove corollary 2.3 using these two results.

**Lemma F.1.** *Let the likelihood $p(y|z)$ be in exponential family (eq. (8)) and prior $p(z) = s(z|\xi_0)$ be in the corresponding conjugate family (eq. (9)). Let $\mathcal{D}$ be a multiset of training data, $\mathcal{D}^*$ a multiset of test data, and let $R_1$ be the Monte Carlo estimator for the PPD with exact inference (eq. (2) with $K = 1$). Let $h(\mathcal{D}^*) = \prod_{y \in \mathcal{D}^*} h(y)$. Then,*

$$\mathbb{E}[R_1]^c = h(\mathcal{D}^*)^c \frac{\exp B(\mathcal{D} + c\mathcal{D}^*)}{\exp B(\mathcal{D})}, \tag{138}$$

*where $c$ is a non-negative integer and $B$ is as in eq. (9).*

*Proof.* Starting from the definition of $R_1$ we have,

$$\mathbb{E}[R_1^c] = \mathbb{E}\left[(p(\mathcal{D}^*|z))^c\right] = \mathbb{E}\left[\left(\prod_{y \in \mathcal{D}^*} p(y|z)\right)^c\right] \tag{139}$$

$$= \mathbb{E}\left[\left(\prod_{y \in \mathcal{D}^*} h(y) \exp\left(T(y)^\top \phi(z) - A(z)\right)\right)^c\right] \tag{140}$$

$$\overset{(a)}{=} \mathbb{E}\left[\left(h(\mathcal{D}^*) \exp\left(T(\mathcal{D}^*)^\top \phi(z) - |\mathcal{D}^*|A(z)\right)\right)^c\right], \tag{141}$$

$$\tag{142}$$

where (a) follows from $T(\mathcal{D}^*) = \sum_{y \in \mathcal{D}^*} T(y)$ and $h(\mathcal{D}^*) = \prod_{y \in \mathcal{D}^*} h(y)$. Doing some basic manipulations, we get

$$\mathbb{E}\left[\left(h(\mathcal{D}^*) \exp\left(T(\mathcal{D}^*)^\top \phi(z) - |\mathcal{D}^*|A(z)\right)\right)^c\right] \tag{143}$$

$$= h(\mathcal{D}^*)^c \mathbb{E}\left[\exp\left(cT(\mathcal{D}^*)^\top \phi(z) - c|\mathcal{D}^*|A(z)\right)\right] \tag{144}$$

$$\overset{(b)}{=} h(\mathcal{D}^*)^c \int \exp\left(cT(\mathcal{D}^*)^\top \phi(z) - c|\mathcal{D}^*|A(z)\right) s(z|\xi_\mathcal{D}) dz \tag{145}$$

$$\overset{(c)}{=} h(\mathcal{D}^*)^c \frac{\int \exp\left(cT(\mathcal{D}^*)^\top \phi(z) - c|\mathcal{D}^*|A(z)\right) \exp\left(T(\mathcal{D})^\top \phi(z) - |\mathcal{D}|A(z)\right) dz}{\exp(B(\xi_\mathcal{D}))} \tag{146}$$

$$\overset{(d)}{=} h(\mathcal{D}^*)^c \frac{\int \exp\left(T(\mathcal{D} + c\mathcal{D}^*)^\top \phi(z) - (|\mathcal{D} + c\mathcal{D}^*|)A(z)\right) dz}{\exp(B(\xi_\mathcal{D}))} \tag{147}$$

$$\overset{(e)}{=} h(\mathcal{D}^*)^c \frac{\exp(B(\xi_{\mathcal{D}+c\mathcal{D}^*}))}{\exp(B(\xi_\mathcal{D}))} \tag{148}$$

$$\tag{149}$$

where (b) and (c) follow from the definition of $s(z|\xi_\mathcal{D})$ (eq. (9)) and the fact that the expectation is under the posterior; (d) follows from the multiset notation (Costa, 2021); (e) follows from the definition of $B$ in eq. (9). $\qquad \square$

**Lemma F.2.** *In a canonical exponential family $p(x|\eta) = h(x) \exp\left(T(x)^\top \eta - A(\eta)\right)$, the looseness of Jensen's equality applied to the log-partition function $A$ at points $v, w$, and $u = \frac{v+w}{2}$ is*

$$\frac{1}{2}(A(v) + A(w)) - A(u) = \frac{1}{2}KL\left(p(x|u) \parallel p(x|v)\right) + \frac{1}{2}KL\left(p(x|u) \parallel p(x|w)\right).$$

*Proof.* The KL-divergence between two canonical exponential family distributions with parameters $v$ and $w$ is given by

$$\text{KL}\left(p(x|w) \| p(x|v)\right) = \underset{p(x|w)}{\mathbb{E}} \log \frac{p(x|w)}{p(x|v)} = \underset{p(x|w)}{\mathbb{E}} \left(T(x)^\top w - T(x)^\top v - A(w) + A(v)\right) \tag{150}$$

$$= (w-v)^\top \underset{p(x|w)}{\mathbb{E}} [T(x)] - A(w) + A(v) \tag{151}$$

$$\overset{(a)}{=} (w-v)^\top \nabla A(w) - A(w) + A(v), \tag{152}$$

where (a) follows from the definition of the gradient of $A$.

Now, rearranging terms in eq. (152) gives an expression for log-partition function $A$ at any point $w$ in terms of the log-partition function $A$ at any other point $v$ and the KL-divergence between the two distributions:

$$A(w) = A(v) + (w-v)^\top \nabla A(w) - \text{KL}\left(p(x|w) \| p(x|v)\right) \tag{153}$$

Replacing $w$ with $u$ in eq. (153), gives

$$A(u) = A(v) + (u-v)^\top \nabla A(u) - \text{KL}\left(p(x|u) \| p(x|v)\right), \tag{154}$$

and replacing $w$ with $u$ and $v$ with $w$ in eq. (153) gives

$$A(u) = A(w) + (u-w)^\top \nabla A(u) - \text{KL}\left(p(x|u) \| p(x|w)\right). \tag{155}$$

On averaging eq. (154) and eq. (155) the $\nabla A(u)$ terms cancel out, and we get

$$A(u) = \frac{1}{2}\left(A(v) + A(w)\right)$$
$$- \frac{1}{2}\text{KL}\left(p(x|u) \| p(x|v)\right) - \frac{1}{2}\text{KL}\left(p(x|u) \| p(x|w)\right) \tag{156}$$

Finally, rearranging the terms, proves the result:

$$\frac{1}{2}\left(A(v) + A(w)\right) - A(u) = \frac{1}{2}\left(\text{KL}\left(p(x|u) \| p(x|v)\right) + \text{KL}\left(p(x|u) \| p(x|w)\right)\right). \tag{157}$$

$\square$

**Theorem F.3** (Repeated). *Take a model with a likelihood $p(y|z)$ in an exponential family (eq. (8)) and a prior $p(z) = s(z|\xi_0)$ in the corresponding conjugate family (eq. (9)). Let $\mathcal{D}^*$ be some test data. Let $R_K$ be the Monte Carlo estimator for the PPD under exact inference (eq. (2)). Then, the signal-to-noise ratio is $SNR(R_K) = \sqrt{K}/\sqrt{\exp(\delta)^2 - 1}$ for*

$$\delta = \frac{1}{2}KL\left(s(z|\mathcal{D} + \mathcal{D}^*) \| s(z|\mathcal{D})\right) + \frac{1}{2}KL\left(s(z|\mathcal{D} + \mathcal{D}^*) \| s(z|\mathcal{D} + 2\mathcal{D}^*)\right) \tag{158}$$

$$= \frac{B(\xi_\mathcal{D}) + B(\xi_{\mathcal{D}+2\mathcal{D}^*})}{2} - B(\xi_{\mathcal{D}+\mathcal{D}^*}), \tag{159}$$

*where for any dataset $\mathcal{D}$, $\xi_\mathcal{D}$ are the parameters that make the conjugate family $s(z|\xi_\mathcal{D})$ equal to the posterior density $p(z|\mathcal{D})$ (eq. (10)), and $B$ is as in eq. (9).*

*Proof.* From Lemma C.1 we get $\text{SNR}(R_K) = \frac{\sqrt{K}}{\sqrt{\exp(\delta)^2 - 1}}$ for $\delta = \frac{1}{2}\log(\mathbb{E}[R_1^2]/\mathbb{E}[R_1]^2)$. Using Lemma F.1, for $c = 1$ and $c = 2$, we can simplify $\delta$ as

$$\delta = \frac{1}{2}\log \frac{\mathbb{E}[R_1^2]}{\mathbb{E}[R_1]^2} = \frac{1}{2}\log \mathbb{E}\left[R_1^2\right] - \log \mathbb{E}\left[R_1\right] \tag{160}$$

$$\overset{(a)}{=} \frac{1}{2}\log h(\mathcal{D}^*)^2 \frac{\exp B(\mathcal{D} + 2\mathcal{D}^*)}{\exp B(\mathcal{D})} - \log h(\mathcal{D}^*) \frac{\exp B(\mathcal{D} + \mathcal{D}^*)}{\exp B(\mathcal{D})} \tag{161}$$

$$\overset{(b)}{=} \frac{1}{2}\log \frac{\exp B(\mathcal{D} + 2\mathcal{D}^*)}{\exp B(\mathcal{D})} - \log \frac{\exp B(\mathcal{D} + \mathcal{D}^*)}{\exp B(\mathcal{D})} \tag{162}$$

$$\overset{(c)}{=} \frac{B(\xi_{\mathcal{D}+2\mathcal{D}^*}) + B(\xi_\mathcal{D})}{2} - B(\xi_{\mathcal{D}+\mathcal{D}^*}), \tag{163}$$

where (a) follows from Lemma F.1 for $c = 1$ and $c = 2$, (b) follows from cancellations of $\log h(\mathcal{D}^*)$, and (c) follows from simple algebra.

Now, observe $B$ in eq. (9) is the log-partition function of a canonical exponential family. Using Lemma F.2, and plugging $v = \xi_{\mathcal{D}}$, $u = \xi_{\mathcal{D}+\mathcal{D}^*}$, and $w = \xi_{\mathcal{D}+2\mathcal{D}^*}$ for conjugate prior family gives eq. (11). $\qquad\square$

# G. Proof for Theorem 3.1

This section contains the proof for theorem 3.1. The idea of the proof is similar to the proof of theorem 2.1 in section C. Definition G.1 defines the log constant and the posterior distribution for the augmented posterior with test data. Lemma G.2 provides the expression for the moments of the naive MC estimator under the approximate inference. Lemma G.3 provides the expression for the KL divergence between two augmented posteriors. Lemma G.4 provides the expression for the sum of KL divergences between three augmented posterior distributions, where the first distribution is the same for the two KL terms. Corollary G.5 simplifies the expression for the sum of KL divergences when the datasets have a specific structure. Finally, we use these four results to prove theorem 3.1.

**Definition G.1.** Let $p(\mathcal{D}|z)$ be the likelihood and $p(z)$ be the prior distribution. Let $q_\mathcal{D}(z)$ be the variational distribution. Let $\mathcal{D}^*$ be some test data. Then,

$$Z_\mathcal{D}(\mathcal{D}^*) := \log \int p(\mathcal{D}^*|z)q_\mathcal{D}(z)dz \qquad \text{and} \qquad q_\mathcal{D}(z|\mathcal{D}^*) := \frac{p(\mathcal{D}^*|z)q_\mathcal{D}(z)}{Z_\mathcal{D}(\mathcal{D}^*)} \tag{164}$$

**Lemma G.2.** *Let $p(\mathcal{D}|z)$ be the likelihood and $p(z)$ be the prior distribution. Let $q_\mathcal{D}(z)$ be the variational distribution. Let $\mathcal{D}^*$ be some test data. Let $p(\mathcal{D} + \mathcal{D}^*|z) = p(\mathcal{D}|z)p(\mathcal{D}^*|z)$ for any datasets $\mathcal{D}$ and $\mathcal{D}^*$. Let $R_K$ be the Monte Carlo estimator for the $PPD_q$ under approximate inference (eq. (2) with $K = 1$). Then,*

$$\mathbb{E}\left[R_1^c\right] = \exp Z_\mathcal{D}(c\mathcal{D}^*), \tag{165}$$

*where $c$ is a non-negative integer.*

*Proof.* The proof is straightforward for $c = 0$ as $Z_\mathcal{D}(\emptyset) = \log \int q_\mathcal{D}(z)dz = 0$. For $c \geq 1$, we have

$$\mathbb{E}\left[R_1^c\right] = \mathbb{E}\left[p(\mathcal{D}^*|z)^c\right] \tag{166}$$

$$= \mathbb{E}\left[p(c\mathcal{D}^*|z)\right] \tag{167}$$

$$= \int p(c\mathcal{D}^*|z)q_\mathcal{D}(z)dz \tag{168}$$

$$= \exp Z_\mathcal{D}(c\mathcal{D}^*). \tag{169}$$

$\square$

**Lemma G.3.** *Let $p(\mathcal{D}|z)$, $p(z)$, and $q_\mathcal{D}(z)$ be as in definition G.1. Let $\mathcal{D}_a$ and $\mathcal{D}_b$ be the three multisets of data. Then,*

$$KL\left(q_\mathcal{D}(z|\mathcal{D}_a) \parallel q_\mathcal{D}(z|\mathcal{D}_b)\right) = \mathbb{E}\left[\log p(\mathcal{D}_a|z) - \log p(\mathcal{D}_b|z)\right] - Z_\mathcal{D}(\mathcal{D}_a) + Z_\mathcal{D}(\mathcal{D}_b) \tag{170}$$

$$\tag{171}$$

*Proof.*

$$\text{KL}\left(q_\mathcal{D}(z|\mathcal{D}_a) \parallel q_\mathcal{D}(z|\mathcal{D}_b)\right) \tag{172}$$

$$= \mathbb{E}\left[\log \frac{\frac{p(\mathcal{D}_a|z)q_\mathcal{D}(z)}{\exp Z_\mathcal{D}(\mathcal{D}_a)}}{\frac{p(\mathcal{D}_b|z)q_\mathcal{D}(z)}{\exp Z_\mathcal{D}(\mathcal{D}_b)}}\right] \tag{173}$$

$$= \mathbb{E}\left[\log \frac{p(\mathcal{D}_a|z)}{p(\mathcal{D}_b|z)}\right] - \log \frac{\exp Z_\mathcal{D}(\mathcal{D}_a)}{\exp Z_\mathcal{D}(\mathcal{D}_b)} \tag{174}$$

$$= \mathbb{E}\left[\log \frac{p(\mathcal{D}_a|z)}{p(\mathcal{D}_b|z)}\right] - Z_\mathcal{D}(\mathcal{D}_a) + Z_\mathcal{D}(\mathcal{D}_b) \tag{175}$$

$$\tag{176}$$

$\square$

**Lemma G.4.** *Let $p(\mathcal{D}|z)$, $p(z)$, and $q_\mathcal{D}(z)$ be as in definition G.1. Let $\mathcal{D}_1$, $\mathcal{D}_2$, and $\mathcal{D}_3$ be the three multisets of data. Let $\mathcal{D}_1$, $\mathcal{D}_2$, and $\mathcal{D}_3$ be the three multisets of data. Then,*

$$\frac{1}{2}KL\left(q_\mathcal{D}(z|\mathcal{D}_3) \,\|\, q_\mathcal{D}(z|\mathcal{D}^*_1)\right) + \frac{1}{2}KL\left(q_\mathcal{D}(z|\mathcal{D}_3) \,\|\, q_\mathcal{D}(z|\mathcal{D}^*_2)\right) \tag{177}$$

$$= \mathbb{E}\left[\log p(\mathcal{D}_3|z) - \frac{1}{2}\log p(\mathcal{D}_1|z) - \frac{1}{2}\log p(\mathcal{D}_2|z)\right] \tag{178}$$

$$+ \frac{Z_\mathcal{D}(\mathcal{D}_1) + Z_\mathcal{D}(\mathcal{D}_2)}{2} - Z_\mathcal{D}(\mathcal{D}_3). \tag{179}$$

*Proof.* Applying the lemma G.3 to $\mathcal{D}_3$ and $\mathcal{D}_1$ gives

$$KL\left(q_\mathcal{D}(z|\mathcal{D}_3) \,\|\, q_\mathcal{D}(z|\mathcal{D}^*_1)\right) \tag{180}$$

$$= \mathbb{E}\left[\log p(\mathcal{D}_3|z) - \log p(\mathcal{D}_1|z)\right] - Z_\mathcal{D}(\mathcal{D}_3) + Z_\mathcal{D}(\mathcal{D}_1) \tag{181}$$

$$\tag{182}$$

and applying it to $\mathcal{D}_3$ and $\mathcal{D}_2$ gives

$$KL\left(q_\mathcal{D}(z|\mathcal{D}_3) \,\|\, q_\mathcal{D}(z|\mathcal{D}^*_2)\right) \tag{183}$$

$$= \mathbb{E}\left[\log p(\mathcal{D}_3|z) - \log p(\mathcal{D}_2|z)\right] - Z_\mathcal{D}(\mathcal{D}_3) + Z_\mathcal{D}(\mathcal{D}_2). \tag{184}$$

$$\tag{185}$$

Now, multiplying the above two equations by $\frac{1}{2}$ and adding them gives

$$\frac{1}{2}KL\left(q_\mathcal{D}(z|\mathcal{D}_3) \,\|\, q_\mathcal{D}(z|\mathcal{D}^*_1)\right) \tag{186}$$

$$+ \frac{1}{2}KL\left(q_\mathcal{D}(z|\mathcal{D}_3) \,\|\, q_\mathcal{D}(z|\mathcal{D}^*_2)\right) \tag{187}$$

$$= \mathbb{E}\left[\log p(\mathcal{D}_3|z) - \frac{1}{2}\log p(\mathcal{D}_1|z) - \frac{1}{2}\log p(\mathcal{D}_2|z)\right] \tag{188}$$

$$+ \frac{Z_\mathcal{D}(\mathcal{D}_1) + Z_\mathcal{D}(\mathcal{D}_2)}{2} - Z_\mathcal{D}(\mathcal{D}_3). \tag{189}$$

$$\square$$

**Corollary G.5.** *Let $p(\mathcal{D}|z)$, $p(z)$, and $q_\mathcal{D}(z)$ be as in definition G.1. Let $\mathcal{D}_1 = c_a\mathcal{D}$, $\mathcal{D}_2 = c_a\mathcal{D} + 2c_b\mathcal{D}^*$, and $\mathcal{D}_3 = c_a\mathcal{D} + c_b\mathcal{D}^*$ be the three multisets of data where $c_a$ and $c_b$ are non-negative integers. Then,*

$$\frac{1}{2}KL\left(q_\mathcal{D}(z|\mathcal{D}_3) \,\|\, q_\mathcal{D}(z|\mathcal{D}^*_1)\right) + \frac{1}{2}KL\left(q_\mathcal{D}(z|\mathcal{D}_3) \,\|\, q_\mathcal{D}(z|\mathcal{D}^*_2)\right) \tag{190}$$

$$= \frac{Z_\mathcal{D}(\mathcal{D}_1) + Z_\mathcal{D}(\mathcal{D}_2)}{2} - Z_\mathcal{D}(\mathcal{D}_3). \tag{191}$$

**Theorem G.6** (Repeated for convenience). *Let $p(\mathcal{D}|z)$ be the likelihood and $p(z)$ be the prior distribution. Let $q_\mathcal{D}(z)$ be the variational distribution. Let $\mathcal{D}^*$ be some test data. Let $p(\mathcal{D} + \mathcal{D}^*|z) = p(\mathcal{D}|z)p(\mathcal{D}^*|z)$ for any datasets $\mathcal{D}$ and $\mathcal{D}^*$. Let $R_K$ be the Monte Carlo estimator for the $PPD_q$ under approximate inference (eq. (2) with $K = 1$). Then, the signal-to-noise ratio of $R_K$ is given by $SNR\left(R_K\right) = \sqrt{K}/\sqrt{\exp(\delta)^2 - 1}$ where*

$$\delta = \frac{1}{2}KL\left(q_\mathcal{D}(z|\mathcal{D}^*) \,\|\, q_\mathcal{D}(z)\right) + \frac{1}{2}KL\left(q_\mathcal{D}(z|\mathcal{D}^*) \,\|\, q_\mathcal{D}(z|2\mathcal{D}^*)\right) \tag{192}$$

$$= \frac{1}{2}Z_\mathcal{D}(2\mathcal{D}^*) - Z_\mathcal{D}(\mathcal{D}^*) \tag{193}$$

*where $Z_\mathcal{D}$ and $q_\mathcal{D}(z|\mathcal{D}^*)$ are as in definition G.1.*

*Proof.*

$$\delta \overset{(a)}{=} \frac{1}{2} \log \frac{\mathbb{E}[R_1^2]}{\mathbb{E}[R_1]^2} \tag{194}$$

$$= \frac{1}{2} \log \mathbb{E}\left[R_1^2\right] - \log \mathbb{E}\left[R_1\right] \tag{195}$$

$$\overset{(b)}{=} \frac{1}{2} Z_{\mathcal{D}}(2\mathcal{D}^*) - Z_{\mathcal{D}}(\mathcal{D}^*) \tag{196}$$

Where $(a)$ follows from lemma C.1 and $(b)$ follows from lemma G.2. Lastly, plugging $\mathcal{D}_1 = \emptyset$ and $\mathcal{D}_2 = 2\mathcal{D}^*$ and $\mathcal{D}_3 = \mathcal{D}^*$ into corollary G.5 and observing $Z_{\mathcal{D}}(\emptyset) = 0$ gives the result. □

# H. Proof for Corollary 3.2

This section contains the proof for corollary 3.2. The proof follow the same structure as the proof for theorem 2.1 in section C. Lemma H.1 provides the expression for the moments of the naive MC estimator under the approximate inference in the conjugate exponential family models. We use these results to prove corollary 3.2.

**Lemma H.1.** *Let the likelihood $p(y|z)$ be as in eq. (8) and a prior $p(z) = s(z|\xi_0)$ be as in eq. (9). Let $q_{\mathcal{D}}(z) = s(z|\eta)$ be in the conjugate family (eq. (9)). Let $\mathcal{D}^*$ be some test data and let $R_1$ be the Monte Carlo estimator for the $PPD_q$ under approximate inference (eq. (2) with $K = 1$). Then,*

$$\mathbb{E}[R_1^c] = h(\mathcal{D}^*)^c \exp\left(B\left(\eta + U(c\mathcal{D}^*)\right) - B\left(\eta\right)\right), \tag{197}$$

*$c$ is a non-negative integer, $B$ is as in eq. (9), and $U(c\mathcal{D}) = c \begin{bmatrix} T(\mathcal{D}) \\ |\mathcal{D}| \end{bmatrix}$ for any dataset $\mathcal{D}$.*

*Proof.* Starting from the definition of $R_{q,1}$ we have,

$$\mathbb{E}[R_{q,1}^c] = \mathbb{E}\left[(p(\mathcal{D}^*|z))^c\right] = \mathbb{E}\left[\left(\prod_{y\in\mathcal{D}^*} p(y|z)\right)^c\right] \tag{198}$$

$$= \mathbb{E}\left[\left(\prod_{y\in\mathcal{D}^*} h(y)\exp\left(T(y)^\top \phi(z) - A(z)\right)\right)^c\right] \tag{199}$$

$$\overset{(a)}{=} \mathbb{E}\left[\left(h(\mathcal{D}^*)\exp\left(T(\mathcal{D}^*)^\top \phi(z) - |\mathcal{D}^*|A(z)\right)\right)^c\right], \tag{200}$$

where (a) follows from $T(\mathcal{D}^*) = \sum_{y\in\mathcal{D}^*} T(y)$ and $h(\mathcal{D}^*) = \prod_{y\in\mathcal{D}^*} h(y)$. Doing some basic manipulations, we get

$$\mathbb{E}\left[\left(h(\mathcal{D}^*)\exp\left(T(\mathcal{D}^*)^\top \phi(z) - |\mathcal{D}^*|A(z)\right)\right)^c\right] \tag{201}$$

$$= h(\mathcal{D}^*)^c \mathbb{E}\left[\exp\left(cT(\mathcal{D}^*)^\top \phi(z) - c|\mathcal{D}^*|A(z)\right)\right] \tag{202}$$

$$\overset{(b)}{=} h(\mathcal{D}^*)^c \mathbb{E}\left[\exp\left(c\left(\begin{bmatrix} T(\mathcal{D}^*) \\ |\mathcal{D}^*| \end{bmatrix}\right)^\top \begin{bmatrix} \phi(z) \\ -A(z) \end{bmatrix}\right)\right] \tag{203}$$

$$\overset{(c)}{=} h(\mathcal{D}^*)^c \mathbb{E}\left[\exp\left(U(c\mathcal{D}^*)^\top \begin{bmatrix} \phi(z) \\ -A(z) \end{bmatrix}\right)\right] \tag{204}$$

$$\overset{(d)}{=} h(\mathcal{D}^*)^c \int \exp\left(U(c\mathcal{D}^*)^\top \begin{bmatrix} \phi(z) \\ -A(z) \end{bmatrix}\right) s(z|\eta)dz \tag{205}$$

$$\overset{(e)}{=} h(\mathcal{D}^*)^c \frac{\int \exp\left(U(c\mathcal{D}^*)^\top \begin{bmatrix} \phi(z) \\ -A(z) \end{bmatrix}\right)\exp\left(\eta^\top \begin{bmatrix} \phi(z) \\ -A(z) \end{bmatrix}\right) dz}{\exp B(\eta)} \tag{206}$$

$$\overset{(f)}{=} h(\mathcal{D}^*)^c \frac{\int \exp\left((U(c\mathcal{D}^*) + \eta)^\top \begin{bmatrix} \phi(z) \\ -A(z) \end{bmatrix}\right) dz}{\exp B(\eta)} \tag{207}$$

$$\overset{(g)}{=} h(\mathcal{D}^*)^c \frac{\exp(B(\eta + U(c\mathcal{D}^*)))}{\exp(B(\eta))} \tag{208}$$

$$= h(\mathcal{D}^*)^c \exp(B(\eta + U(c\mathcal{D}^*)) - B(\eta)) \tag{209}$$

where (b) just collects the terms in the exponent into a single vector; (c) defines $U(c\mathcal{D}) = c \begin{bmatrix} T(\mathcal{D}) \\ |\mathcal{D}| \end{bmatrix}$ for any dataset $\mathcal{D}$; (d) and (e) follows as expectation is under the variational distribution and the definition of conjugate family in eq. (9); (f) follows from some simple algebra; (g) follows from the definition of $B$ in eq. (9). $\qquad\square$

**Theorem H.2.** *Take a model with a likelihood $p(y|z)$ in an exponential family (eq. (8)) and a prior $p(z) = s(z|\xi_0)$ in the corresponding conjugate family (eq. (9)). Let $q_{\mathcal{D}}(z) = s(z|\eta)$ be an approximate distribution in the corresponding*

*conjugate family (eq. (9)) with parameters $\eta$. Let $\mathcal{D}^*$ be a multiset of test data and let $R_{1,q}$ be the Monte Carlo estimator for the $PPD_q$ (eq. (2) with $K = 1$). Then, the signal-to-noise ratio is $SNR(R_{1,q}) = \frac{1}{\sqrt{\exp(\delta)^2 - 1}}$ for*

$$\delta = \frac{1}{2} KL\left(s(z|\eta + U(\mathcal{D}^*)) \parallel s(z|\eta)\right) + \frac{1}{2} KL\left(s(z|\eta + U(\mathcal{D}^*)) \parallel s(z|\eta + U(2\mathcal{D}^*))\right) \tag{210}$$

$$= \frac{B(\eta) + B(\eta + U(2\mathcal{D}^*))}{2} - B(\eta + U(\mathcal{D}^*)), \tag{211}$$

*where $B$ is as in eq. (9) and $U(c\mathcal{D}) = c\begin{bmatrix} T(\mathcal{D}) \\ |\mathcal{D}| \end{bmatrix}$ for any dataset $\mathcal{D}$ and non-negative integer $c$.*

*Proof.* From Lemma C.1 we get $SNR(R_K) = \frac{\sqrt{K}}{\sqrt{\exp(\delta)^2 - 1}}$ for $\delta = \frac{1}{2}\log(\mathbb{E}[R_1^2]/\mathbb{E}[R_1]^2)$. Then

$$\delta = \frac{1}{2}\log\frac{\mathbb{E}[R_1^2]}{\mathbb{E}[R_1]^2} = \frac{1}{2}\log\mathbb{E}\left[R_1^2\right] - \log\mathbb{E}\left[R_1\right] \tag{212}$$

$$\stackrel{(a)}{=} \frac{1}{2}\left(B(\eta + U(2\mathcal{D}^*)) - B(\eta)\right) - \left(B(\eta + U(\mathcal{D}^*)) - B(\eta)\right) \tag{213}$$

$$\stackrel{(b)}{=} \frac{B(\eta + U(2\mathcal{D}^*)) + B(\eta)}{2} - B(\eta + U(\mathcal{D}^*)), \tag{214}$$

where (a) follows from Lemma H.1 for $c = 1$ and $c = 2$ and cancellations of $\log h(\mathcal{D}^*)$ terms and (b) form simple algebraic manipulations.

Now, observe $B$ in eq. (9) is the log-partition function of a canonical exponential family. Using Lemma F.2, and plugging $v = \eta$, $u = \eta + U(\mathcal{D}^*)$, and $w = \eta + U(2\mathcal{D}^*)$ for conjugate prior family gives the eq. (15). $\square$

# I. General experimental details

All our code is implemented in JAX (Bradbury et al., 2018) and run on a single NVIDIA A100 GPU. In table 6, we provide the expressions for computation of different metrics from the results in tables 1 to 3 and section 5.4.

**Note on BBVI.** We rely on using standard BBVI techniques for most of our experiments. The hope of BBVI is to allow practitioners to not worry about designing special approximation families for each model $p(\mathcal{D}, z)$ (Ranganath et al., 2014; Kucukelbir et al., 2017; Agrawal et al., 2020; Ambrogioni et al., 2021a;b; Burroni et al., 2024). Instead, BBVI treats models as black boxes—only requiring access to $\nabla_z \log p(\mathcal{D}, z)$ to update the variational parameters using the stochastic gradients of a variational objective (for instance, IW-ELBO). Ongoing research in BBVI focuses on automating other algorithmic choices (Kucukelbir et al., 2017; Agrawal et al., 2020; Ambrogioni et al., 2021a;b; Burroni et al., 2024). Such optimization schemes greatly improve the applicability of BBVI and come pre-implemented in popular probabilistic programming languages like Pyro (Bingham et al., 2019), NumPyro (Phan et al., 2019), and Stan (Carpenter et al., 2017). While we implement our own inference schemes for this paper, we expect the results to be similar if we use the aforementioned libraries.

Table 6: Summary of the expressions of metrics and their computations for the table Tables 1 to 4. We report SNR $(R)$ in terms of $\mathbb{E}[R]$ and $\mathbb{V}[R]$ and report explicit form in tables 7 and 8. We use $S = 1000$ for all our experiments. The results are then averaged over ten independent trials to generate mean and standard deviation numbers in tables 1 to 4

| Expression | Computation | | Expression | Computation |
|---|---|---|---|---|
| $\mathbb{E}[\log R_K]$ | $z_{s,k} \sim q_{\mathcal{D}}, \frac{1}{S} \sum_{s=1}^{S} \left[ \log \frac{1}{K} \sum_{k=1}^{K} p(\mathcal{D}^*\|z_{s,k}) \right]$ | | SNR $(R_K)$ | $\mathbb{E}[R_K] \Big/ \sqrt{\mathbb{V}[R_K]}$ |
| $\mathbb{E}[\log R_K^{\mathrm{IS}}]$ | $z_{s,k} \sim r_w, \frac{1}{S} \sum_{s=1}^{S} \left[ \log \frac{1}{K} \sum_{k=1}^{K} \frac{p(\mathcal{D}^*\|z_{s,k})q(z_{s,k}\|\mathcal{D})}{r_w(z_{s,k})} \right]$ | | SNR $\left(R_K^{\mathrm{IS}}\right)$ | $\mathbb{E}[R_K^{\mathrm{IS}}] \Big/ \sqrt{\mathbb{V}[R_K^{\mathrm{IS}}]}$ |

Table 7: Mean of SNR for different estimators.

| Expression | Computation |
|---|---|
| $\mathbb{E}[R_K]$ | $z_{s,k} \sim q_{\mathcal{D}}, \frac{1}{S} \sum_{s=1}^{S} \left[ \frac{1}{K} \sum_{k=1}^{K} p(\mathcal{D}^*\|z_{s,k}) \right]$ |
| $\mathbb{E}[R_K^{\mathrm{IS}}]$ | $z_{s,k} \sim r_w, \frac{1}{S} \sum_{s=1}^{S} \left[ \frac{1}{K} \sum_{k=1}^{K} \frac{p(\mathcal{D}^*\|z_{s,k})q(z_{s,k}\|\mathcal{D})}{r_w(z_{s,k})} \right]$ |

Table 8: Variance of SNR for different estimators.

| Expression | Computation |
|---|---|
| $\mathbb{V}[R_K]$ | $z_{s,k} \sim q_{\mathcal{D}}, \frac{1}{S-1} \sum_{s=1}^{S} \left[ \frac{1}{K} \sum_{k=1}^{K} p(\mathcal{D}^*\|z_{s,k}) - \mathbb{E}[R_K] \right]^2$ |
| $\mathbb{V}[R_K^{\mathrm{IS}}]$ | $z_{s,k} \sim r_w, \frac{1}{S-1} \sum_{s=1}^{S} \left[ \frac{1}{K} \sum_{k=1}^{K} \frac{p(\mathcal{D}^*\|z_{s,k})q(z_{s,k}\|\mathcal{D})}{r_w(z_{s,k})} - \mathbb{E}[R_K^{\mathrm{IS}}] \right]^2$ |

## J. Exponential Family models: Additional Details

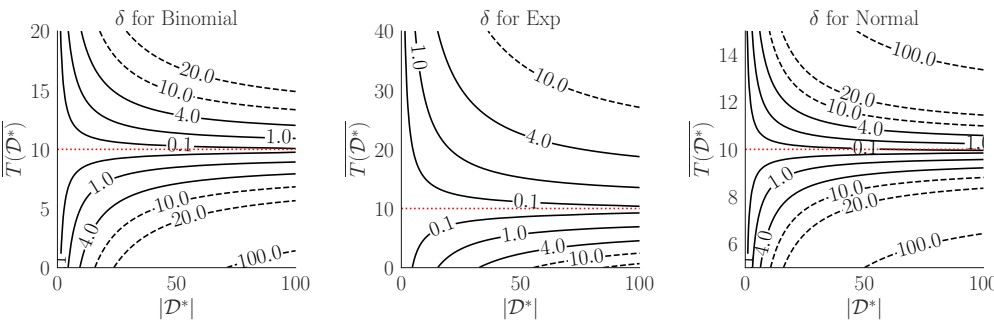

Figure 9: $\delta$ **contours.** Setting exactly the same as Figure 5

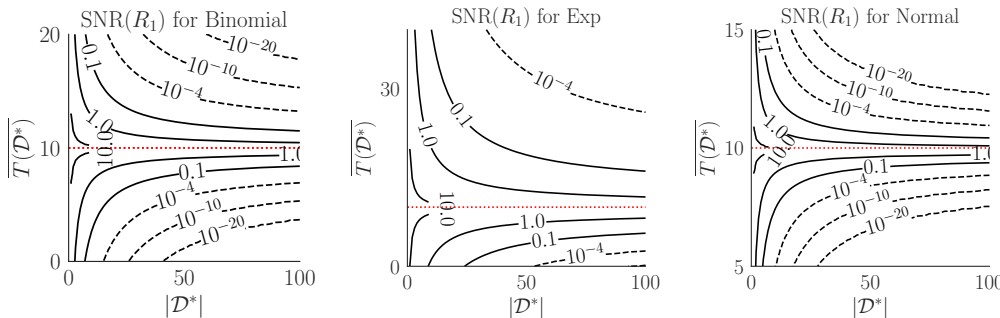

Figure 10: **SNR contours.** (Repeated for easier reference). Setting is the same as the Figure 5.

For each of the three models, we fix the number of training data points $|\mathcal{D}| = 100$ and number of test data points $|\mathcal{D}^*| = 100$. Then, to sample the training data such that the mean statistics of the data $\overline{T(\mathcal{D})} \approx 10$, we sample from the likelihood distributions by carefully adjusting the parameters. This means, for normal we sample from $\mathcal{N}(10, 1)$; for Exp we sample from $\mathrm{Exp}(0.1)$; and for Binomial we sample from $\mathrm{Binomial}(100, 0.1)$.

Then, to sample the test data, we first select the region of high $\delta$ from the Figure 5 and then roughly try to match the target mean statistics by carefully adjusting the parameters. For Normal, we sample from $\mathcal{N}(5, 1)$ to target $\overline{T(\mathcal{D}^*)} \approx 5$; for Exp we sample from $\mathrm{Exp}(0.025)$ to target $\overline{T(\mathcal{D}^*)} \approx 40$; and for Binomial we sample from $\mathrm{Binomial}(100, 0.4)$ to target $\overline{T(\mathcal{D}^*)} \approx 40$. This strategy leads to the numbers in table 5. Note, we only use one test and train setting for our experiments. The results reported in tables 1 and 2 are averaged our ten independent estimations for a single data setting.

Table 9: For the three models: Normal, Exp, and Binomial, we identify the exponential family form from Section 2. For likelihood in eq. (8), we identify base measure $h(y)$, one-to-one parameter mapping $\phi(z)$, and log-partition function $A(z)$. Note, the sufficient statistics $T(y) = y$ for all models. For the conjugate prior in eq. (9), we identify the log partition function $B(\xi)$, where $\xi = (\xi_T, \xi_n)^\top$.

| Model | $p(y\|z)$ | $h(y)$ | $\phi(z)$ | $A(z)$ | $B(\xi)$ |
|---|---|---|---|---|---|
| Normal | $\mathcal{N}(y\|z, \sigma^2)$ | $\frac{\exp(-\frac{y^2}{2\sigma^2})}{\sqrt{2\pi\sigma^2}}$ | $\frac{z}{\sigma^2}$ | $\frac{z^2}{2\sigma^2}$ | $\frac{1}{2}\left[\log\frac{2\pi\sigma^2}{\xi_n} + \frac{\xi_T^2}{\sigma^2\xi_n}\right]$ |
| Exp | $\mathrm{Exp}(y\|z)$ | $1$ | $-z$ | $-\log z$ | $\log\frac{\Gamma(\xi_n+1)}{\xi_T^{\xi_n+1}}$ |
| Binomial | $\mathrm{Bin}(y\|n, z)$ | $\binom{n}{y}$ | $\log\frac{z}{1-z}$ | $-n\log(1-z)$ | $\log\frac{\Gamma(\xi_T+1)\Gamma(n\xi_n-\xi_T+1)}{\Gamma(n\xi_n+2)}$ |

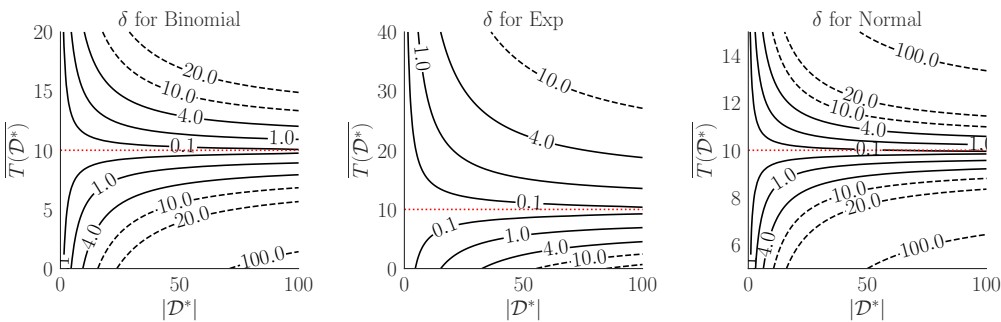

Figure 11: (Repeated for easier reference). $\delta$ **contours.** Settings exactly the same as Figure 5.

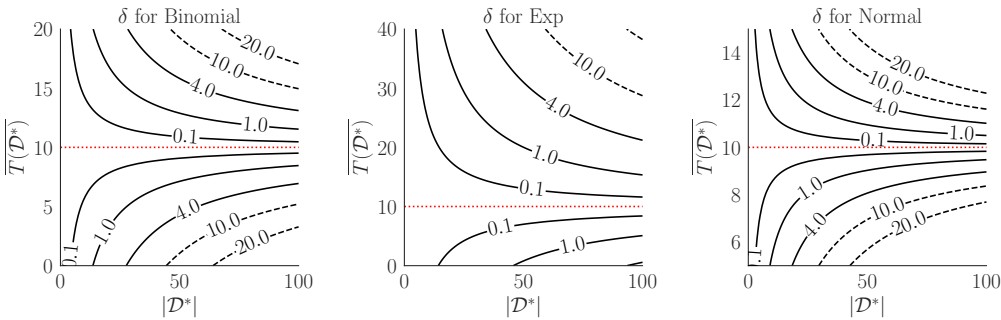

Figure 12: $\delta$ **contours.** For each model, we first fix the training data set such that $\overline{T(\mathcal{D})} = 10$ (shown with red dotted line) and $|\mathcal{D}| = 1000$. For all the models, increasing the number of training data points results in lower $\delta$ for a given test data statistics when compared to Figure 11.

We learn a Gaussian variational approximation for each of the three models from Table 2. For the models with constrained latent variables (Exponential and Binomial), we transform $z$ to an unconstrained space and then adjust the logarithm of the determinant of the Jacobian for correct density estimation (please, see (Kucukelbir et al., 2017, Section 2.3) for more details on such transformations). Our variational family has two unconstrained parameters: $\mu$ and $\sigma$. To ensure positivity of standard deviation, we transform $\sigma$ with the soft-plus function.

We consider two options to initialize $\mu$ and $\sigma$: Laplace's approximation and standard Normal. To pick from the two options, we evaluate ELBO using 1000 samples and chose the option with higher ELBO value. For Laplace's approximation, we use JAX's BFGS optimizer (Bradbury et al., 2018) (for each model, BFGS took less than 50 estimations of $\log p(z, \mathcal{D})$).

To learn the variational parameters, we optimize standard ELBO using ADAM (Kingma & Ba, 2015) with a learning rate of $0.001$ for $10,000$ iterations. For each iteration, we use a batch of 16 samples for estimating the DReG gradient (Tucker et al., 2019).

We learn a parameterized Gaussian proposal distribution for each of the three models from Tables 1 and 2. For the models with constrained latent variables (Exponential and Binomial), we transform $z$ to an unconstrained space and then adjust the logarithm of the determinant of the Jacobian for correct density estimation (please, see (Kucukelbir et al., 2017, Section 2.3) for more details on such transformations). Our parameterized proposal distribution has two unconstrained parameters: $\mu$ and $\sigma$. To ensure positivity of standard deviation, we transform $\sigma$ with the soft-plus function.

We consider two options to initialize $\mu$ and $\sigma$: Laplace's approximation and standard Normal. To pick from the two options, we evaluate IW-ELBO$_M$ using 1000 samples and chose the option with higher IW-ELBO$_M$ value. For Laplace's approximation, we use JAX's BFGS optimizer (Bradbury et al., 2018) (for each model, BFGS took less than 50 estimations of $\log p(\mathcal{D}^*|z)p(z|\mathcal{D})$ or $p(\mathcal{D}^*|z)q_{\mathcal{D}}(z)$).

To learn the proposal parameters, we optimize IW-ELBO$_M$ using ADAM (Kingma & Ba, 2015) with a learning rate of

0.001 for 1000 iterations. For each iteration, we use a single sample of the DReG estimator. Note, a single sample of DReG estimator for IW-ELBO$_M$ uses $M$ samples. We set $M = 16$ for all our experiments. Note, even after counting the Laplace's approximation estimations, we use less than $20,000$ estimations of $\log p(\mathcal{D}^*|z)p(z|\mathcal{D})$ for learning the proposal.

**Fatter tails.** For the Binomial model, we observe that the estimates for PPD$_q$ are higher than the estimates for PPD$_q$. This can be explained from two observations. First, the approximate posterior has fatter right tails than the true posterior, and second, the test data mean lies to the right of the training data mean (see table 5). This means that the approximate posterior places more mass in the region of test data and the PPD$_q$ will be higher than PPD$_q$. In fig. 13, we plot the densities for the exact posterior and the learned approximation $q_\mathcal{D}$. We also plot an inset-zoomed-in version to highlight the fatter right tail of the approximate posterior. Remember, the variational approximation in the constrained space is obtained after transforming the unconstrained Gaussian variational approximation.

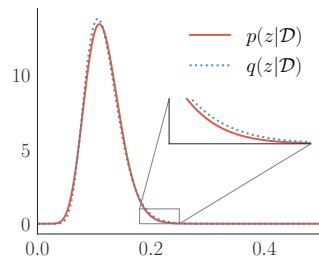

Figure 13: Fatter right tail of $q_\mathcal{D}$ for Binomial model.

### J.1. Empirical Validation for eq. (5)

We consider a model similar to the Normal model where likelihood $p(y|z)$ is given by a multivariate normal $\mathcal{N}(y|z, \Sigma)$ with unknown mean $z \in \mathbb{R}^d$ and known variance $\Sigma = \mathbb{I}_d$. A multivariate Normal prior $\mathcal{N}(z|0, \mathbb{I}_d)$ gives a conjugate model as in section 2. For this model, we vary the number of latent dimensions $d \in \{1, 10, 100, 10000, 10000\}$. For each $d$, we create a training data set $\mathcal{D}$ with 1000 data points, and set test data $\mathcal{D}^*$ to $\mathcal{D}$, that is, the mean statistics for training and test data sets match exactly. In fig. 14, we plot the $\delta$ from the approximation in eq. (5) (shown in blue dotted lines with crosses), and compare it against the $\delta$ from exact calculations in eq. (3) (shown in red solid lines with dots). The approximation is accurate for all $d$, and $\delta$ scales linearly as predicted. This means for higher dimensional latent spaces, we can have extremely low SNR $(R_1)$ even if test data statistics match exactly to the training data statistics.

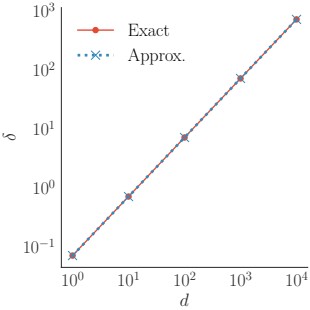

Figure 14: $\delta$ from approximation in eq. (5) (blue dotted line) is accurate when compared to $\delta$ from exact expression in eq. (3) (red solid lines). Also, $\delta$ scales linearly with $d$ (eq. (5)).

### J.2. Effect of increasing the number of training data points

In fig. 12, we consider the effect of increasing the number of training data points from $|\mathcal{D}| = 100$ (fig. 11), to $|\mathcal{D}| = 1000$ while holding the mean training statistics, $T(\mathcal{D}) = 10$, the same. As the number of training data points increases, $\delta$ gets smaller for any given test setting. To understand why, note $\delta$ as in eq. (3) involves two KL divergences: one between posteriors $p(z|\mathcal{D} + \mathcal{D}^*)$ and $p(z|\mathcal{D})$, and the other between posteriors $p(z|\mathcal{D} + \mathcal{D}^*)$ and $p(z|\mathcal{D} + 2\mathcal{D}^*)$. Intuitively, as the number of training data points increases, we either require more test data or bigger mismatch between test data and training data for the two KL divergences to be large. Thus, for any given test data setting, we expect $\delta$ to be smaller as the number of training data points increases.

# K. Linear Regression: Additional Details

## K.1. Experimental Details

We consider the exact inference settings and start with a baseline scenario where none of the three factors influencing SNR are too high. Thereafter, we independently increase the three factors: mismatch, the dimensionality of the latent space, and the size of the test data to create three additional scenarios. We use the standard normal prior and likelihood with $\sigma^2 = 1$.

**Baseline.** We set the number of training data points to 1000, the dimensionality of latent space $d = 10$, and the number of mismatched copies $m = 1$. We then forward sample a training data set $\mathcal{D}$ and then generate the mismatched data $\mathcal{D}_\Delta$ by adding a mismatch vector $\Delta = 2$ to the response vector $y_\mathcal{D}$.

**More mismatch.** We keep the training data same as in the baseline scenario and increase the mismatch vector to $\Delta = 10$.

**More test data.** We keep the training data same as in the baseline scenario and increase the number of mismatched copies to $m = 10$.

**More dimensions.** We keep the number of training data points, the number of mismatched copies, and the mismatch vector same as in the baseline scenario and increase the dimensionality of the latent space to $d = 100$. We forward sample the training data set $\mathcal{D}$ and then generate the mismatched data $\mathcal{D}_\Delta$ by adding a mismatch vector $\Delta = 2$ to the response vector $y_\mathcal{D}$.

Figure 8 reports the results from estimating $\text{PPD}_q$ using simple MC estimator $R_K$ from eq. (2) for $K = 10^0, 10^1, \ldots, 10^6$. The error bands are the five and ninety-five percentile intervals based on 1000 independent estimations.

For LIS, we learn a full-rank Gaussian proposal distribution by optimizing the IW-ELBO from eq. (18) with $M = 16$ using the DReG estimator and ADAM optimizer with a learning rate of 0.001 for 1000 iterations. We consider different initialization techniques for the variational parameters: Laplace's approximation and standard Normal, and pick the one that provides higher initial ELBO. For each optimization step, we use 8 copies to average the IW-ELBO gradient. For LIS, we learn the proposal once, and do $1,000$ independent estimations to estimate the error bands.

# L. Logistic Regression: Additional Details

**Baseline.** We set the number of training data points to 1000, the dimensionality of latent space $d = 10$, and the number of mismatched copies $m = 1$. We forward sample a training data set $\mathcal{D}$ and then generate the mismatched data $\mathcal{D}_\Delta$ by flipping the first $\Delta = 0.1$ fraction of the response vector $y_\mathcal{D}$.

**More mismatch.** We keep the training data same as in the baseline scenario and increase the mismatch fraction to $\Delta = 1.0$.

**More test data.** We keep the training data same as in the baseline scenario and increase the number of mismatched copies to $m = 10$.

**More dimensions.** We keep the number of training data points, the number of mismatched copies, and the mismatch fraction same as in the baseline scenario and increase the dimensionality of the latent space to $d = 100$. We forward sample the training data set $\mathcal{D}$ and then generate the mismatched data $\mathcal{D}_\Delta$ by flipping the first $\Delta = 0.1$ fraction of the response vector $y_\mathcal{D}$.

We learn a full-rank Gaussian variational approximation by optimizing the standard ELBO objective using the ADAM optimizer with a learning rate of 0.001 for 1000 iterations. We consider different initialization techniques for the variational parameters: Laplace's approximation and standard Normal, and pick the one that provides higher initial ELBO. For each optimization step, we use 16 independent copies to average the ELBO gradient.

For LIS, we learn a full-rank Gaussian proposal distribution by optimizing the IW-ELBO from eq. (18) with $M = 16$ using the ADAM optimizer with a learning rate of 0.001 for 1000 iterations. We consider different initialization techniques for the variational parameters: Laplace's approximation and standard Normal, and pick the one that provides higher initial ELBO. We use eight independent copies to average the gradient of IW-ELBO.

# M. Hierarchical Model: Additional Details

We use MovieLens25M (Harper & Konstan, 2015), a dataset of 25 million ratings for over 60,000 movies, rated by more than 160,000 users, and use movie features made of tag relevance scores collected by Vig et al. (2012)).

Movielens25M originally uses a 5 point ratings system. To get binary ratings, we map ratings greater than 3 points to 1 and less than and equal to 3 to 0. We pre-process the data to drop users with more than 1,000 ratings—leaving around 20M ratings. Also, we PCA the movie features to reduce their dimensionality to 10. We used a train-test split such that, for each user, one-tenth of the ratings are in the test set. This gives us $\approx$ 18M ratings for training (and $\approx$ 2M ratings for testing). Our of these we randomly select 100 users for experiments.

For Gaussian VI, we use a full-rank Gaussian. We optimize standard ELBO using ADAM for 1000 iterations with step-size of 0.001. For each optimization step, we use 16 copies to average the gradient.

For flow VI, we use a real-NVP flow with 10 coupling layers for all our experiments. We define each coupling layer to be comprised of two transitions, where a single transition corresponds to affine transformation of one part of the latent variables. For example, if the input variable for the $k^{th}$ layer is $z^{(k)}$, then first transition is defined as

$$z_{1:d} = z_{1:d}^{(k)}$$
$$z_{d+1:D} = z_{d+1:D}^{(k)} \odot \exp\left(s_k^a(z_{1:d}^{(k)})\right) + t_k^a(z_{1:d}^{(k)}). \tag{215}$$

where, for the function $s$ and $t$, super-script $a$ denotes first transition and sub-script $k$ denotes the $k^{th}$ layer. For the next transition, the $z_{d+1:D}$ part is kept unchanged and $z_{1:d}$ is affine transformed similarly to obtain the layer output $z^{(k+1)}$ (this time using $s_k^b(z_{d+1:D}^{(k)})$ and $t_k^b(z_{d+1:D}^{(k)})$). This is also referred to as the alternating first half binary mask. Both, scale($s$) and translation($t$) functions of single transition are parameterized by the same fully connected neural network(FNN). More specifically, for first transition in above example, a single FNN takes $z_{1:d}^{(k)}$ as input and outputs both $s_k^a(z_{1:d}^{(k)})$ and $t_k^a(z_{1:d}^{(k)})$. Thus, the skeleton of the FNN, in terms of the size of the layers, is as $[d, H, H, 2(D - d)]$ where, $H$ denotes the size of the two hidden layers ($H$=32 for all our experiments).

The hidden layers of FNN use a leaky rectified linear unit with slope = 0.01, while the output layer uses a hyperbolic tangent for $s$ and remains linear for $t$. We initialize the parameters of the neural networks from normal distribution $\mathcal{N}(0, 0.001^2)$. This choice approximates standard normal initialization. We optimize standard ELBO with sticking the landing (STL) (Roeder et al., 2017) gradient using ADAM for 1000 iterations with step-size of 0.001. For each optimization step, we use 16 copies to average the gradient.

To learn the proposal distribution for the learned IS estimator, we use a real-NVP flow with architecture described above. We initialize it with parameters from the variational distribution. For the Gaussian VI, we fix the base distribution for the flow to the variational distribution. For flow VI, we use the same architecture for the proposal distribution and simply initialize using the parameters of the variational distribution. We optimize IW-ELBO with DReG estimator using ADAM for 100 iterations with step-size of 0.001. For each optimization step, we use 8 copies to average the gradient.

Table 10: Wall-clock times for one trial of the MovieLens experiments (times were stable across trials with minor variations).

| Method | No. of samples $(K)$ | Time for Simple MC (s) | Time for IS (s) | Time for learning the proposal (s) |
|---|---|---|---|---|
| Flow VI | $K = 10^3$ | 8.82 | 17.11 | 33.76 |
| | $K = 10^6$ | 4487.01 | 6781.45 | 28.06 |
| Gaussian VI | $K = 10^3$ | 7.15 | 20.13 | 34.29 |
| | $K = 10^6$ | 4094.95 | 6611.28 | 33.98 |

**Running time.** Our method has some overhead to optimize the IW-ELBO (independent of the number of samples $K$) and some overhead to perform importance weighting on each sample. When the SNR is low, naive Monte Carlo sampling might be faster. However, when a large number of samples is needed, the overhead to optimize the IW-ELBO can be negligible, and the per-sample overhead is small compared to the benefits. For instance, table 10 plots the time for MovieLens experiments. If you look at Table 4, switching from naive Monte Carlo from $10^3$ to $10^6$ samples (with $1,000\times$ the computational cost) improves SNR less than switching to importance sampling with $10^3$ samples (with around $1.5\times$ the cost).

