# OpenReview forum: "Understanding the difficulties of posterior predictive estimation"
_ICML.cc/2025/Conference — ICML 2025 poster_

### Official Review · Reviewer_hMnG · 2025-03-09

**Overall Recommendation:** 3

**Summary:**

The authors study the problem of Monte Carlo estimation of the density of the posterior predictive distribution for (approximate) Bayesian inference. They show that simple Monte Carlo estimation can have a low signal to noise ratio (SNR) if the training data and test data are substantially different, the dimension of the problem is high or the test set is much larger than the training set. They propose an importance sampling approach based on learning a sampling distribution using an Importance Weighted Evidence lower bound, and provide empirical support for this method at least partially mitigating the observed problems with low signal to noise ratio when using simple Monte Carlo estimation to estimate the (predictive) posterior density.

**Claims And Evidence:**

*Claim: Bias in estimating the log predictive density can lead to unreliable comparisons between methods.*

Figure 1 illustrates the bias in estimating the log predictive density. While bias is clearly illustrated, it would be more compelling if the authors provided an example where the bias leads to issues in comparison between the methods. In the current example, it seems like we would reach the same conclusion, regardless of how many samples are used in estimation. And it seems reasonable to believe this is the correct conclusion (that flow VI is leading to higher predictive density). Also the standard error bars shown are disjoint for all numbers of samples, so it really seems comparison in this case is reliable, even if estimation is poor.

*Claim: SNR is low when training and test data are 'mismatched'*
Figures 2 and 8 show simulations supporting this claim. The linear regression case also provides support for this claim.

*Claim: SNR decreases as the dimension of the latent space increases*
Figure 2 and figure 8 (right) shows a simulation supporting this claim. Equation 5 provides a heuristic argument for this claim. I found the sketch of the heuristic argument in the main text incomplete. In particular, it wasn't clear to me what was meant by "the corresponding Bayesian CLT approximations". The link to section D provided didn't link to the appendix. When I went to section D of the appendix, I expected to see a clear statement of what is intended by A1 (i.e. what Baysian CLT is being used in this heuristic, along with a citation). But I couldn't find this. Please provide additional detail or point me to where it is already in the text. I strongly suspect some assumptions on the data-generating process for both test and train locations are needed for equation 70 (in particular, something like that all the data is sampled iid from some distribution). I also find assumption A2 difficult to understand at an intuitive level. Could you give concrete examples where it holds/does not hold? Currently think the argument is so heuristic that it does not provide significant insight beyond the simulations.  The linear regression case provides some support for this claim in a particular concrete (although somewhat limited) setting.

*Claim: SNR is low when the test set is large compared to the test set*
Figure 2 and 8 show simulations supporting this claim. The heuristic support for this claim follows the same argument as for the claim about dimension, and so my earlier comments apply. The linear regression case provides some support for this claim in a particular concrete setting.

*Claim: Adaptive importance sampling can increase SNR relative to simple Monte Carlo*

Tables 1,2 and 3 as well as figure 8 support this claim.

**Essential References Not Discussed:**

I did not see significant gaps in discussion of related literature (Appendix B). However, I am not familiar enough with several areas related to this paper in order to feel confident nothing has been missed.

**Experimental Designs Or Analyses:**

The simulation studies are reasonable illustrations of the arguments made in the paper. I looked over the model details provided in section 7, and did not have significant concerns.

**Methods And Evaluation Criteria:**

The authors evaluate standard Monte Carlo and the proposed importance sampling in terms of expected log predictive density and estimated signal to noise ratio. While there is not a ground truth expected log predictive density available in many of the experiments, this metric is still useful as 1.) it is commonly used in practice for model comparison between methods, and so the quality of its estimation is of interest. 2.) the direction of bias in estimating this quantity is known, and so higher values must represent lower amounts of bias. The signal to noise ratio also seems a useful metric to report since it is the focus of most of the discussion of the paper.

**Other Comments Or Suggestions:**

The proof sketch for Theorem 2.1 repeatedly refers to steps as *simple*. I don't think this is useful. Either the reader will understand how this step is done and agree it is simple, or they will not in which case telling them it is simple is not helpful (and discouraging).

appendix A.3, typo: “margina”

Punctuation is missing in many equations in the appendix (e.g. eqn 34, 39, 88, 91, 107, 113). Eqns 40, 108 shouldn't be there as it is empty (likely results from an additional new line command).

**Other Strengths And Weaknesses:**

Weakness: Some sections of the text did not have clear takeaways. For example, the discussion of approximate inference (section 3) gave formulas for the SNR, but didn't provide insight into when I should expect the SNR to be high or low with approximate inference. In the discussion for exponential families, there is a condition discussed in terms of statistics lying on a particular ray, but I didn't see how this can be related back to quantities related to the analysis (the available data, the prior and likelihood, the approximate inference performed).

**Questions For Authors:**

What is the main (practical) insight I am supposed to take from section 2.2? I didn't see how the description around "looseness" in Jensen's inequality supported the main claims of the paper, nor how to turn it into practical insights about when I would expect the SNR to be low/high.

**Relation To Broader Scientific Literature:**

To the best of my knowledge, there is not a significant amount of literature studying the quality of Monte Carlo estimation of the (approximate) posterior based on independent samples. The proposed method in the paper builds on ideas from importance weighted variational inference.

**Theoretical Claims:**

I looked over the appendices for the proof of theorems 2.1 and 2.2. I the argument structure makes sense and I did not see errors. Overall the theoretical claims seem believable, although in certain places heuristic arguments made are imprecise (see details in other boxes).

---

> ### Author Rebuttal · Authors · 2025-04-01
>
> We thank the reviewer for their detailed comments. We will address the typos, equation formatting, and other minor suggestions as is and offer comments to other concerns below.
>
> >... Bayesian CLT approximations ... provide additional detail ...
>
> While there are several versions of Bernstein-von Mises, we believe Theorem 10.1 from Vaart (1998) suffices for our use case. Note that Theorem 10.1 states the result in terms of the true parameters. For the extension to the maximum likelihood estimate, see the discussion between Lemma 10.3 and Lemma 10.4.
>
> In short, we assume that the posterior $p(z \vert \mathcal D) \approx \mathcal N(z_{\textrm{MLE}}, \frac{1}{|\mathcal{D}|} I^{-1}(z_{\textrm{MLE}})),$ where $I$  is the estimate of the Fisher information matrix evaluated at the maximum likelihood estimate; such that,  $I^{-1} (z_{\textrm{MLE}}) = -(\frac{1}{|\mathcal D|} \nabla^2_z \log p(\mathcal D \vert z))^{-1}$. Since the size of the dataset cancels out, we get the expressions in equations 65, 66, and 67. In the revision, we will add the discussion on Bernstein-von Mises and the above clarification after Corollary D.2.
>
> [1] Vaart, A. W. van der. "Bayes Procedures." _Asymptotic Statistics_. Cambridge: Cambridge University Press, 1998.
>
> >Assumptions on the data-generating process for ...  equation 70.
>
> In spirit, this assumption is not that different from assuming the data-generating distributions are the same. The quantity in eq. 70 ($\frac{1}{|\mathcal D|}S_{\mathcal D}^{-1}$) is the estimate of the Fisher information matrix (FIM) under different datasets. We essentially assume that the train and test data sets are similar enough such that estimates (MLE and FIM) under these datasets will be the same. As the reviewer notices, it is tempting to only assume that the train and test data distributions are similar. However, even after assuming the same distribution, we will require additional assumptions for a result in terms of estimates under two given datasets. This assumption about the estimates is precisely the reason we (very explicitly) keep the result informal. We will add this discussion after eq. 70, and if the reviewers suggest, we will reword the assumption in terms of FIM.
>
> >What is being defined in definition C.2?
>
> There is no specific assumption on the form of the posterior. We are essentially writing Bayes' rule. However, we define $V$ explicitly as we use it as a function of data $\mathcal D$ in later expressions. We made an organizational choice to aid the reader, but we can move it to an inline definition if preferred.
>
> >The discussion of approximate inference ... but didn't provide insight ...
>
> The main insight is that for approximate posteriors that closely resemble the true posterior, we expect the relationships from Section 2 to hold as is. However, for arbitrary distributions,  it is hard to make a precise statement beyond "SNR depends on how much the *posterior* $q_\mathcal{D}(z|\mathcal{D}^∗)$ varies from the *prior* $q_\mathcal{D}(z)$." Intuitively, the training data only enters these equations through the conditioning, or, equivalently, from the approximate posterior $q_\mathcal D(z)$. Since arbitrary approximate posteriors may have arbitrary relationships with training data, it is not immediately obvious how to make a more precise statements. We hope that future research can explore this further. We will add this discussion at the end of Section 3 and in the limitations paragraph in Section 6.
>
> >What is the main (practical) insight I am supposed to take from section 2.2?
>
> The analysis in Section 2.2 is complementary to the main claims of the paper. From the earlier parts in Section 2, we know that when the training and test datasets match and are large enough, the posteriors in eq. 3 are similar, and the SNR is high. The same is implied by the two conditions at the end of section 2.2. When $\xi_\mathcal D$ is large, the training dataset will be large (note that $\xi_\mathcal D$ includes both the statistics and the number of data points, see eq. 10). When the points $\xi_\mathcal D$ and $\xi_{\mathcal D + \mathcal D^*}$ lie close to the ray emanating from the origin, the datasets have to be similar. Combined together, these two conditions also support the view that when the datasets are large and similar, we expect the SNR to be high. We favored the illustration in Figure 6 in place of words to derive the main takeaway. However, we will add some of this explicit discussion to the main text.
>
> Overall, we thank the reviewer for these insightful comments. We strongly believe that the concerns raised by them will make the paper stronger. We look forward to addressing any remaining concerns. Otherwise, we feel confident in our abilities to add the requested explanatory text, and hope the reviewer will consider a strong recommendation for our work.

---

> > ### Comment · Reviewer_hMnG · 2025-04-03
> >
> > Thanks to the authors for the response. I think the paper raises interesting points. I have decided to maintain my score.
> >
> > I appreciate the authors additional descriptions of take-aways of the main results from the paper (regarding approximate inference and section 2.2).
> >
> > Section 2.2: I still don't entirely see the new insight gained from section 2.2 that wasn't already conveyed in the general case. Is it that it suffices, for datasets to be "similar" in the exponential family case, it suffices that the resulting sufficient statistics are similar? The geometric insight from working an exponential family seems nice, but I'm still not clear what I should take away from the section practically, that I would not have concluded already from earlier sections. It seems from the authors response that section 2.2 is meant as further support of the main claims. Mostly, I'd like to see a clear discussion of how the exponential family case relates to the general case, and what structure is particular to the exponential family case that adds additional insight into the problem considered by the authors.

---

> > > ### Author Response · Authors · 2025-04-03
> > >
> > > We truly appreciate the reviewer for their additional time and offer more explanations below.
> > >
> > > We begin with the friendly thought that "insight, to some degree, is in the eye of the beholder!" We acknowledge that we did not clearly explain what insight section 2.2 offers. Roughly, we think of corollary 2.2 as saying, "SNR is determined by Jensen's inequality applied to the log-partition function of the exponential family." We find this insightful because log-partition functions are, in some sense, "soft-max" functions or (very informally) "rounded cones." In particular, we know that they are "rounded" (causing looseness and thus low SNR) near the origin, but if you follow any ray away from the origin, they eventually become "flat" (causing tightness and thus high SNR).
> > >
> > > We think this provides some further insight, but don’t necessarily necessarily mean that these insights leads to practical changes to practical algorithmic choices. (Of course, if you truly had a conjugate family, the posterior would be known in closed form.) We find log-partition view of the problem to be an appealing explanation for "why" the SNR is high or low: The impact of both differences in sizes of datasets and differences in observed moments on SNR can be understood in terms of how they position points along the surface of the "cone" given by the log-partition function—small datasets position you near the origin where the log partition function is rounded, while different moments mean the points do not lie on a ray pointing near the origin (as shown in Figure 6)
> > >
> > > The corollary 2.2 is also used to compute SNR in closed form for exponential families, which is useful for making plots. Table 9 in Appendix J provides the expressions for the log partition function. These expressions are plugged into eq. 12 to calculate SNR and generate Figure 5 and Figures 9-12. These visualizations provide tangible evidence of how SNR can quickly decay in relatively simple scenarios.
> > >
> > > We hope these explanations better portray our insights, and we look forward to addressing any remaining concerns.

---

### Official Review · Reviewer_qCWu · 2025-03-14

**Overall Recommendation:** 4

**Summary:**

This paper addresses the issue of unreliable posterior predictive density (PPD) estimates when using a simple Monte Carlo (MC) approach, highlighting the previously under-recognized issue of low signal-to-noise ratio (SNR). The authors theoretically analyze and empirically demonstrate that the SNR for posterior predictive estimation significantly decreases under three conditions: increasing mismatch between training and test data, increasing dimensionality of the latent variable space, and an increasing relative size of the test dataset. They propose an adaptive importance sampling method—learned IS—based on maximizing a variational proxy for SNR, to mitigate this issue, showing substantial improvement in predictive estimation accuracy.

**Claims And Evidence:**

The main claim in this work is the signal-to-noise ratio (SNR) of the simple MC estimator can sometimes be extremely low, leading to unreliable estimates. To demonstrate this, the authors provide a theoretical analysis in section 2. They provide an analytical expression for SNR in Theorem 2.1 and they demonstrate it empirically on several models: linear regression, logistic regression, and hierarchical models.

**Essential References Not Discussed:**

The authors appear to have cited the relevant works adequately. No essential references seem missing.

**Experimental Designs Or Analyses:**

The soundness of the experimental designs was checked, particularly the experiments with linear regression, logistic regression, exponential family models, and hierarchical models. The setups appear valid, and the experimental results strongly support the theoretical analyses. The use of illustrative examples clearly demonstrates the issues with low SNR and the effectiveness of the learned IS method.

**Methods And Evaluation Criteria:**

The methods proposed in this paper are appropriate for the problem at hand and follow obviously from the analysis presented. Specifically, the authors identify the central issue of low signal-to-noise ratio (SNR) in simple Monte Carlo estimations of posterior predictive densities (PPDs). To mitigate this, they propose an adaptive importance sampling method, named "Learned Importance Sampling" (LIS). In this approach, they optimize a variational proxy to the SNR by maximizing an importance-weighted evidence lower-bound (IW-ELBO), to learn a more efficient proposal distribution.

The methodological choice is well-justified, as direct optimization of the optimal proposal distribution (which would maximize SNR directly) is often intractable. Hence, optimizing a proxy measure like the IW-ELBO, which has clear theoretical connections to minimizing estimator variance, is a reasonable and effective strategy (Section 4)​
.
The authors also empirically demonstrate that this method significantly improves estimator accuracy across various scenarios, including exponential family models, linear regression, logistic regression, and hierarchical models, providing robust evidence that their methodological choice is practically effective and relevant for the stated problem.

**Other Comments Or Suggestions:**

No

**Other Strengths And Weaknesses:**

Strengths:
1. Clearly identifies and thoroughly analyzes a subtle but critical issue.
2. Provides theoretical grounding and extensive empirical validation.
3. Introduces a simple yet effective methodological improvement.

Weaknesses:
Computational complexity and scalability of the learned IS method, particularly in very high-dimensional or large-scale applications, may be a concern but it is less important since the contributions are primarily theoretical.

**Questions For Authors:**

No

**Relation To Broader Scientific Literature:**

The paper's contributions lie within the broader literature of approximate inference and Monte Carlo estimation. It builds upon work on importance sampling and also related to prior research on variational inference, predictive uncertainty, and Bayesian model evaluation.

**Theoretical Claims:**

I reviewed the correctness of the theoretical derivations provided, particularly Theorems 2.1 and 2.2. The proofs provided appear to be sound and accurately support the claims regarding the decay of SNR under specified conditions.

---

> ### Author Rebuttal · Authors · 2025-04-01
>
> Thank you very much for your detailed and encouraging review. We appreciate your recognition of our theoretical analysis and the empirical validation of our proposed LIS method, as well as your insights into potential computational challenges. Your constructive feedback is truly invaluable.

---

### Official Review · Reviewer_LsR8 · 2025-03-14

**Overall Recommendation:** 4

**Summary:**

This paper provides a theoretical framework explaining the severe signal-to-noise ratio (SNR) degradation observed in naive posterior predictive distribution (PPD) estimators.
It rigorously demonstrate that, even with exact inference, SNR diminishes with increasing: (1) training-test data mismatch, (2) latent space dimensionality, and (3) test data size relative to training data.
These theoretical findings are empirically validated through numerical experiments on both synthetic and real-world datasets.

**Claims And Evidence:**

The paper's claims are rigorously supported by comprehensive references and robust mathematical proofs, built upon well-justified assumptions

**Essential References Not Discussed:**

N/A.

**Experimental Designs Or Analyses:**

The numerical experiments provide clear empirical validation for all theoretical claims presented. A natural and compelling extension would be to investigate whether these theoretical findings hold within the context of Bayesian neural networks, a research area of considerable interest to the contemporary machine learning community.

**Methods And Evaluation Criteria:**

This paper analyzes scenarios involving exact posterior distributions, represented as $Q_{D}(z) = P(Z|D)$, as well as cases where  $P(Z|D)$ is approximated using variational inference (VI) or Laplace approximation. This comprehensive analysis provides a complete theoretical framework.

The theory's reliability is further strengthened by its validation on both synthetic and real-world datasets.

**Other Comments Or Suggestions:**

While the appendix provides a comprehensive review of related work, integrating key discussions into the main text would enhance understanding for readers unfamiliar with the field's cutting-edge developments.

**Other Strengths And Weaknesses:**

N/A.

**Questions For Authors:**

N/A.

**Relation To Broader Scientific Literature:**

This paper identifies three scenarios leading to rapid signal-to-noise ratio (SNR) decay in posterior predictive distributions (PPDs) and proposes adaptive importance sampling as a mitigation strategy, providing valuable insights into PPD failure modes for the machine learning community.

**Theoretical Claims:**

The proofs presented herein are, to the best of my ability, accurate.

---

> ### Author Rebuttal · Authors · 2025-04-01
>
> Thank you so much for your thoughtful and constructive review. We appreciate your recognition of the theoretical contributions and the thoroughness of our analysis. Your feedback is invaluable and greatly encouraging.
>
> For the camera-ready version, we will plan to use the extra space to add more commentary on the takeaways from sections 2.2 and 3 (as pointed out in [response to the reviewer hMnG](https://openreview.net/forum?id=TzfGuKazvf&noteId=ywqVSRwl8H)). We will also use the extra space to move more of the related works discussion to the main text (as indicated in [response to the reviewer CNiq](https://openreview.net/forum?id=TzfGuKazvf&noteId=hM1fsV2z27)). In fact, we are open to moving the model details to the appendix and, instead, moving all of the related works section into the main text. Please let us know if you have a strong preference for one or the other.

---

### Official Review · Reviewer_CNiq · 2025-03-18

**Overall Recommendation:** 2

**Summary:**

This paper provides a theoretical investigation of Monte Carlo estimation of posterior predictive distributions. The signal-to-noise ratio (SNR) of Monte Carlo estimation is shown to decrease under three conditions: increasing mismatch between training and test, increasing dimensionality of latent space, and increasing size of test data. The contributions consist of (1) a theoretical analysis of the SNR in several settings (exact linear regression, exact conjugate exponential family, and approximate inference), and (2) a learned importance sampling procedure designed to improve the quality of the Monte Carlo estimate.

**Claims And Evidence:**

The theoretical claims are supported by proofs in Appendices C - H and Tables 1 - 4 show the improvements due to the learned importance sampling.

**Essential References Not Discussed:**

References discussing e.g. Monte Carlo Variance and variance reduction techniques.

**Experimental Designs Or Analyses:**

The experiments seem to be reasonable but I did not verify them in detail.

**Methods And Evaluation Criteria:**

The experimental setup seems reasonable, consisting of exponential family, linear regression, logistic regression, and a hierarchical model on the MovieLens data. Evaluation criteria include lower bounds on the log posterior predictive density and the signal-to-noise ratio, which make sense for this work.

**Other Comments Or Suggestions:**

p. 2, l. 57: ratio

**Other Strengths And Weaknesses:**

The structure of the paper is nonstandard which makes it more difficult to read. There is no background section building up to the results and justifying the use of SNR to analyze PPD estimation. Related work is contained only in the appendix whereas model details are placed in the main paper after the conclusion. More connections to the literature justifying the use of SNR and arguing for the significance/impact of these results are needed.

**Questions For Authors:**

1. Why is SNR a good metric for analyzing the quality of Monte Carlo estimation of the PPD?
2. How do the theoretical results here influence the way that practitioners estimate PPDs or the way they select approximate posterior distributions?

**Relation To Broader Scientific Literature:**

The results should be better connected to the literature of analyzing the variance of Monte Carlo estimators. In this paper, the signal-to-noise ratio is the object of concern, and to my knowledge this has not been analyzed. The related work section goes back only a decade or so which makes it difficult to assess the significance of the results in this paper.

**Theoretical Claims:**

I did not examine the proofs in detail. The results seem to intuitively correspond to expressions from information geometry.

---

> ### Author Rebuttal · Authors · 2025-04-01
>
> We thank the reviewer for their feedback. We will address the minor concerns and offer some specific comments below.
>
> >Why is SNR a good metric for analyzing the quality of Monte Carlo estimation of the PPD?
>
> We study SNR because it is equivalent to relative variance and bakes in the idea of how large the estimator variance is relative to the target quantity. The idea of relative variance becomes crucial when the target quantity is numerically small, as in the case of $\textrm{PPD}_q$ values. To make this precise, let’s consider an example where the $\log \textrm{PPD}_q= -100$ (for reference, all estimates of $\log \textrm{PPD}_q$ in Tables 1, 2, 3, and 4 are lower than $-100$). Also, consider an unbiased estimator $R$ with variance $\mathbb V(R)=10^{-20}$. In the absolute sense, the variance of this estimator is low; however, $R$ carries more noise than signal. To see why, note that $\textrm{PPD}_q=\exp(-100) \approx 3.72\times10^{-44}.$ Intuitively, $R$ varies on the scale of $10^{-10}$ (standard deviation) and will produce noisy approximations of the target value that is the order of $10^{-44}$. SNR naturally captures this intuition: $\textrm{SNR}(R) = \textrm{PPD}_q/\sqrt{\mathbb V [R]} \approx 3.72\times10^{-34}$ and flags the estimator as poor.
>
> Several works before us have studied the SNR of unbiased estimators of numerically small quantities (as cited in lines 59-61). For the works cited in the main paper, the focus is to study the behavior of gradient estimators since the value of the gradients can become very small as the optimization proceeds. In our extended search, we found two more works that study the SNR of gradient estimators for VI [1, 2] and one more work that studies SNR for the policy gradients in RL [3] (the definition of the SNR in [3] is different from our ratio of moments definition; however, the idea to study the SNR is similarly motivated). In hindsight, we agree this discussion should be more prominent. We will add these citations and the above discussion to the already existing discussion in lines 59-61.
>
> [1] Liévin, Valentin, et al. "Optimal variance control of the score-function gradient estimator for importance-weighted bounds." NeurIPS, 2020.
>
> [2] Rudner, Tim GJ, et al. "On signal-to-noise ratio issues in variational inference for deep Gaussian processes." ICML, 2021.
>
> [3] Roberts, John, and Russ Tedrake. "Signal-to-noise ratio analysis of policy gradient algorithms." NeurIPS, 2008.
>
> >How do the theoretical results here influence the way that practitioners estimate PPDs or the way they select approximate posterior distributions?
>
> We believe our theoretical results are of independent interest and also provide useful insights for practitioners. First, our work serves as a strong cautionary note. We hope that practitioners monitor (and report) the SNR value of their PPD estimates alongside the $\log \textrm{PPD}$ value. Second, whenever the SNR values of the naive estimator are low, our work shows that this does not necessarily reflect on the accuracy of the approximate posterior but simply on our ability to accurately estimate PPD. In cases of low SNR, we suggest practitioners use approaches like LIS to improve the reliability of the estimates and obtain a clearer picture of the relative performance of the different approximate posterior methods.
>
> We briefly touch on the above discussion in the conclusion paragraph in Section 6 and will expand on this in the update.
>
> > Related work is contained only in the appendix, whereas model details are placed in the main paper after the conclusion.
>
> Due to space limitations, we moved the related work section to the appendix. In the camera-ready version, one more page is allowed, and we plan to use it to bring more of the discussion on the related works into the main paper (note that a small paragraph is present in Section 6). If the reviewers believe that the paper will benefit from moving *all* of the related work section to the main paper, then we propose moving the Model Details section to the appendix to create more space.
>
> > The related work section goes back only a decade or so, which makes it difficult to assess the significance of the results in this paper.
>
> Any research review can fall short of being exhaustive. We believe we made an honest attempt to cover the relevant literature. We cite research on PPD evaluation in Section B, and we will further expand on our motivation to study SNR (as mentioned in this rebuttal). If we are missing some relevant work, we will be more than happy to include it. However, we respectfully disagree that it is difficult to judge our work. We will be obliged if the reviewer can point us to specific literature that we have missed, and without which it becomes difficult to assess the significance of our analysis.
>
> Overall, we thank the reviewer for their time and hope that they will reconsider their position in light of the explanations and other reviews.

---

### Decision · Program_Chairs · 2025-05-01

**Decision:**

Accept (poster)

**Comment:**

The paper received positive feedback for its theoretical and empirical analysis with two reviewers recommending acceptance and two leaning toward a weak decision (1 x accept, 1 x reject). While all reviewers acknowledge the novelty and rigor of the theoretical contributions, concerns were raised about novelty, clarity, and the limited discussion of related literature in the main text. The rebuttal addressed several concerns, but some issues remained (in particular the take-away of Section 2.2). All reviewers agree that the paper raises interesting points.